# Principal Component Projection and Regression in Nearly Linear Time through Asymmetric SVRG

**Yujia Jin**
Stanford Universty
yujiajin@stanford.edu

**Aaron Sidford**
Stanford Universty
sidford@stanford.edu

## Abstract

Given a data matrix $\mathbf{A} \in \mathbb{R}^{n \times d}$, *principal component projection (PCP)* and *principal component regression (PCR)*, i.e. projection and regression restricted to the top-eigenspace of $\mathbf{A}$, are fundamental problems in machine learning, optimization, and numerical analysis. In this paper we provide the first algorithms that solve these problems in nearly linear time for fixed eigenvalue distribution and large $n$. This improves upon previous methods which have superlinear running times when both the number of top eigenvalues and inverse gap between eigenspaces is large. We achieve our results by applying rational polynomial approximations to reduce PCP and PCR to solving asymmetric linear systems which we solve by a variant of SVRG. We corroborate these findings with preliminary empirical experiments.

## 1 Introduction

PCA is one of the most fundamental algorithmic tools for analyzing large data sets. Given a data matrix $\mathbf{A} \in \mathbb{R}^{n \times d}$ and a parameter $k$ the classic *principal component analysis* (PCA) problem asks to compute the top $k$ eigenvectors of $\mathbf{A}^\top \mathbf{A}$. This is a core computational task in machine learning and often used for feature selection [1–3], data visualization [4, 5], and model compression [6].

However, as $k$ becomes large, the running time of PCA can degrade. Even just writing down the output takes $\Omega(kd)$ time and the performance of many methods degrade with $k$. This high-computational cost for exploring large-cardinality top-eigenspaces has motivated researchers to consider prominent tasks solved by PCA, for example *principal component projection (PCP)* which asks to project a vector onto the top-$k$ eigenspace, and *principal component regression (PCR)* which asks to solve regression restricted to this top-$k$ eigenspace (see Section 1.2).

Recent work [7, 8] showed that the dependence on $k$ in solving PCP and PCR can be overcome by instead depending on *eigengap* $\gamma$, defined as the ratio between the smallest eigenvalue in the space projected onto and the largest eigenvalue of the space projected orthogonal to. These works replace the typical $\text{poly}(k)\text{nnz}(\mathbf{A})$ dependence in runtime with a $\text{poly}(1/\gamma)\text{nnz}(\mathbf{A})$ (at the cost of lower order terms), by reducing these problems to solving $\text{poly}(1/\gamma)$ ridge-regression problems. Unfortunately, for large-scale problems, as data-set sizes grow so too can $k$ and $1/\gamma$, yielding large super-linear running times for all previously known methods (see Section 1.4). Consequently, this leaves the following fundamental open problem:

*Can we obtain nearly linear running times for solving PCP and PCR to high precision, i.e. with running time $\tilde{O}(\text{nnz}(\mathbf{A}))$ plus an additive term depending only on the eigenvalue distribution?*

The main contribution of the paper is an affirmative answer to this question. We design randomized algorithms that solve PCP and PCR with high probability in nearly linear time. Leveraging rational polynomial approximations we reduce these problems to solving asymmetric linear systems, which

we solve by a technique we call *asymmetric SVRG*. Further, we provide experiments demonstrating the efficacy of this method.

## 1.1 Approach

To obtain our results, critically we depart from the previous frameworks in Frostig et al. [7], Allen-Zhu and Li [8] for solving PCP and PCR. These papers use polynomial approximations to the sign function to reduce PCP and PCR to solving ridge regression. Their runtime is limited by the necessary $\Omega(1/\gamma)$ degree for polynomial approximation of the sign function shown by Eremenko and Yuditskii [9]. Consequently, to obtain nearly linear runtime, a new insight is required.

In our paper, we instead consider rational approximations to the sign function and show that these efficiently reduce PCP and PCR to solving a particular class of squared linear systems. The closed form expression for the best rational approximation to sign function was given by Zolotarev [10] and has recently been proposed for matrix sign approximation [11]. The degree of such rational functions is logarithmic in $1/\gamma$, leading to much fewer linear systems to solve. While the *squared systems* $[(\mathbf{A}^\top \mathbf{A} - c\mathbf{I})^2 + \mu^2 \mathbf{I}]\mathbf{x} = \mathbf{b}, \mu > 0$ induced by this rational approximation are computationally more expensive to solve, as compared with simple ridge regression problems $[\mathbf{A}^\top \mathbf{A} + \mu \mathbf{I}]\mathbf{x} = \mathbf{b}, \mu > 0$, interestingly, we show that these systems can still be solved in nearly linear time for sufficiently large matrices. As a by-product of this analysis, we also obtain an efficient algorithm for leveraging linear system solvers to apply the square-root of a positive semidefinite (PSD) matrix to a vector, where we call a matrix $\mathbf{M}$ positive semidefinite and denote $\mathbf{M} \succeq \mathbf{0}$ when $\forall \mathbf{x}, \mathbf{x}^\top \mathbf{M}\mathbf{x} \geq 0$.

We believe the solver we develop for these squared systems is of independent interest. Our solver is a variant of the stochastic variance-reduced gradient descent algorithm (SVRG) [12] modified to solve asymmetric linear systems. Our iterative method can be viewed as an instance of the variance-reduced algorithm for monotone operators discussed in Section 6 of Palaniappan and Bach [13], with a more careful analysis of the error. We also combine this method with approximate proximal point [14] or Catalyst [15] to obtain accelerated variants.

The conventional wisdom when solving asymmetric systems $\mathbf{M}\mathbf{x} = \mathbf{b}$ that are not positive semidefinite (PSD), i.e. $\mathbf{M} \not\succeq \mathbf{0}$, is to instead solve its PSD counterpart $\mathbf{M}^\top \mathbf{M}\mathbf{x} = \mathbf{M}^\top \mathbf{b}$. However, this operation can greatly impair the performance of stochastic methods, e.g. SVRG [12], SAG [16], etc. (See Section 3.) The solver developed in this paper constitutes one of few known cases where transforming it into asymmetric system solving enables better algorithm design and thus provides large savings (see Corollary 1.) Ultimately, we believe this work on SVRG-based methods outside of convex optimization as well as our improved PCP and PCR algorithms may find further impact.

## 1.2 The Problems

Here we formally define the PCP (Definition 1), PCR (Definition 2), Squared Ridge Regression (Definition 3), and Square-root Computation (Definition 4) problems we consider throughout this paper. Throughout, we let $\mathbf{A} \in \mathbb{R}^{n \times d}$ $(n \geq d)$ denote a data matrix where each row $\mathbf{a}_i \in \mathbb{R}^n$ is viewed as a datapoint. Our algorithms typically manipulate the positive semidefinite (PSD) matrix $\mathbf{A}^\top \mathbf{A}$. We denote the eigenvalues of $\mathbf{A}^\top \mathbf{A}$ as $\lambda_1 \geq \lambda_2 \geq \cdots \geq \lambda_d \geq 0$ and corresponding eigenvectors as $\boldsymbol{\nu}_1, \boldsymbol{\nu}_2, \cdots, \boldsymbol{\nu}_d \in \mathbb{R}^d$, i.e. $\mathbf{A}^\top \mathbf{A} = \mathbf{V}\boldsymbol{\Lambda}\mathbf{V}^\top$ with $\mathbf{V} \stackrel{\text{def}}{=} (\boldsymbol{\nu}_1, \cdots, \boldsymbol{\nu}_d)^\top$ and $\boldsymbol{\Lambda} \stackrel{\text{def}}{=} \mathbf{diag}(\lambda_1, \cdots, \lambda_d)$.

Given eigenvalue threshold $\lambda \in (0, \lambda_1)$ we define $\mathbf{P}_\lambda \stackrel{\text{def}}{=} (\boldsymbol{\nu}_1, \cdots, \boldsymbol{\nu}_k)(\boldsymbol{\nu}_1, \cdots, \boldsymbol{\nu}_k)^\top$ as a projection matrix projecting any vector onto the top-$k$ eigenvectors of $\mathbf{A}^\top \mathbf{A}$, i.e. $\mathrm{span}\{\boldsymbol{\nu}_1, \boldsymbol{\nu}_2, \cdots, \boldsymbol{\nu}_k\}$, where $\lambda_k$ is the minimum eigenvalue of $\mathbf{A}^\top \mathbf{A}$ no smaller than $\lambda$, i.e. $\lambda_k \geq \lambda > \lambda_{k+1}$. Without specification $\|\cdot\|$ is the standard $\ell_2$-norm of vector or matrix.

Given $\gamma \in (0, 1)$, the goal of a PCP algorithm is to project any given vector $\mathbf{v} = \sum_{i \in [d]} \alpha_i \boldsymbol{\nu}_i$ in a desired way: mapping $\boldsymbol{\nu}_i$ of $\mathbf{A}^\top \mathbf{A}$ with eigenvalue $\lambda_i$ in $[\lambda(1+\gamma), \infty)$ to itself, eigenvector $\boldsymbol{\nu}_i$ with eigenvalue $\lambda_i$ in $[0, \lambda(1-\gamma)]$ to $\mathbf{0}$, and any eigenvector $\boldsymbol{\nu}_i$ with eigenvalue $\lambda_i$ in between the gap to anywhere between $\mathbf{0}$ and $\boldsymbol{\nu}_i$. Formally, we define the PCP as follows.

**Definition 1** (Principal Component Projection). *The principal component projection (PCP) of* $\mathbf{v} \in \mathbb{R}^d$ *at threshold* $\lambda$ *is* $\mathbf{v}_\lambda^* = \mathbf{P}_\lambda \mathbf{v}$. *Given threshold* $\lambda$ *and eigengap* $\gamma$, *an algorithm* $\mathcal{A}_{\mathrm{PCP}}(\mathbf{v}, \epsilon, \delta)$ *is an*

$\epsilon$-*approximate PCP algorithm if with probability* $1-\delta$, *its output satisfies following:*

$$\bullet \| \mathbf{P}_{(1+\gamma)\lambda}(\mathcal{A}_{\mathrm{PCP}}(\mathbf{v})-\mathbf{v}) \| \leq \epsilon \|\mathbf{v}\|; \quad and \quad \| (\mathbf{I} - \mathbf{P}_{(1-\gamma)\lambda})\mathcal{A}_{\mathrm{PCP}}(\mathbf{v}) \| \leq \epsilon \|\mathbf{v}\| \quad (1)$$

$$\bullet \| (\mathbf{P}_{(1+\gamma)} - \mathbf{P}_{(1-\gamma)\lambda})(\mathcal{A}_{\mathrm{PCP}}(\mathbf{v})-\mathbf{v}) \| \leq \| (\mathbf{P}_{(1+\gamma)} - \mathbf{P}_{(1-\gamma)\lambda})\mathbf{v} \| + \epsilon \|\mathbf{v}\|$$

The goal of a PCR problem is to solve regression restricted to the particular eigenspace we are projecting onto in PCP. The resulting solution should have no correlation with eigenvectors $\boldsymbol{\nu}_i$ corresponding to $\lambda_i \leq \lambda(1-\gamma)$, while being accurate for $\boldsymbol{\nu}_i$ corresponding to eigenvalues above $\lambda_i \geq \lambda(1+\gamma)$. Also, it shouldn't have too large correlation with $\boldsymbol{\nu}_i$ corresponding to eigenvalues between $(\lambda(1-\gamma), \lambda(1+\gamma))$. Formally, we define the PCR problem as follows.

**Definition 2** (Principal Component Regression). *The principal component regression (PCR) of an arbitrary vector* $\mathbf{b} \in \mathbb{R}^n$ *at threshold* $\lambda$ *is* $\mathbf{x}^*_\lambda = \min_{\mathbf{x} \in \mathbb{R}^d} \|\mathbf{A}\mathbf{P}_\lambda \mathbf{x} - \mathbf{b}\|$. *Given threshold* $\lambda$ *and eigengap* $\gamma$, *an algorithm* $\mathcal{A}_{\mathrm{PCR}}(\mathbf{b}, \epsilon, \delta)$ *is an* $\epsilon$-*approximate PCR algorithm if with probability* $1-\delta$, *its output satisfies following:*

$$\|(\mathbf{I} - \mathbf{P}_{(1-\gamma)\lambda})\mathcal{A}_{\mathrm{PCR}}(\mathbf{b}, \epsilon)\| \leq \epsilon\|\mathbf{b}\| \quad and \quad \|\mathbf{A}\mathcal{A}_{\mathrm{PCR}}(\mathbf{b}, \epsilon) - \mathbf{b}\| \leq \|\mathbf{A}\mathbf{x}^*_{(1+\gamma)\lambda} - \mathbf{b}\| + \epsilon\|\mathbf{b}\| . \quad (2)$$

We reduce PCP and PCR to solving squared linear systems. The solvers we develop for this squared regression problem defined below we believe are of independent interest.

**Definition 3** (Squared Ridge Regression Solver). *Given* $c \in [0, \lambda_1]$, $\mathbf{v} \in \mathbb{R}^d$, *we consider a squared ridge regression problem where exact solution is* $\mathbf{x}^* = ((\mathbf{A}^\top\mathbf{A} - c\mathbf{I})^2 + \mu^2\mathbf{I})^{-1}\mathbf{v}$. *We call an algorithm* RidgeSquare$(\mathbf{A}, c, \mu^2, \mathbf{v}, \epsilon, \delta)$ *an* $\epsilon$-*approximate squared ridge regression solver if with probability* $1-\delta$ *it returns a solution* $\tilde{\mathbf{x}}$ *satisfying* $\|\tilde{\mathbf{x}} - \mathbf{x}^*\| \leq \epsilon\|\mathbf{v}\|$.

Using a similar idea of rational polynomial approximation, we also examine the problem of $\mathbf{M}^{1/2}\mathbf{v}$ for arbitrarily given PSD matrix $\mathbf{M}$ to solving PSD linear systems approximately.

**Definition 4** (Square-root Computation). *Given a PSD matrix* $\mathbf{M} \in \mathbb{R}^{n \times n}$ *such that* $\mu\mathbf{I} \preceq \mathbf{M} \preceq \lambda\mathbf{I}$ *and* $\mathbf{v} \in \mathbb{R}^n$, *an algorithm* SquareRoot$(\mathbf{M}, \mathbf{v}, \epsilon, \delta)$ *is an* $\epsilon$-*approximate square-root solver if with probability* $1-\delta$ *it returns a solution* $\mathbf{x}$ *satisfying* $\|\mathbf{x} - \mathbf{M}^{1/2}\mathbf{v}\| \leq \epsilon\|\mathbf{M}^{1/2}\mathbf{v}\|$.

### 1.3 Our Results

Here we present the main results of our paper, all proved in Appendix D. For data matrix $\mathbf{A} \in \mathbb{R}^{n \times d}$, our running times are presented in terms of the following quantities: Input sparsity $\mathrm{nnz}(\mathbf{A}) \stackrel{\text{def}}{=}$ number of nonzero entries in $\mathbf{A}$; Frobenius norm $\|\mathbf{A}\|_F^2 \stackrel{\text{def}}{=} \mathrm{Tr}(\mathbf{A}^\top\mathbf{A})$; stable rank $\mathrm{sr}(\mathbf{A}) \stackrel{\text{def}}{=} \|\mathbf{A}\|_F^2/\|\mathbf{A}\|_2^2 = \|\mathbf{A}\|_F^2/\lambda_1$; condition number of top-eigenspace: $\kappa \stackrel{\text{def}}{=} \lambda_1/\lambda$. When presenting running times we use $\tilde{O}$ to hide polylogarithmic factors in the input parameters $\lambda_1, \gamma, \mathbf{v}, \mathbf{b}$, error rates $\epsilon$, and success probability $\delta$.

For $\mathbf{A} \in \mathbb{R}^{n \times d}$ ($n \geq d$), $\mathbf{v} \in \mathbb{R}^d$, $\mathbf{b} \in \mathbb{R}^n$, without loss of generality we assume $\lambda_1 \in [1/2, 1]^1$. Given threshold $\lambda \in (0, \lambda_1)$ and eigengap $\gamma \in (0, 2/3]$, the main results of this paper are the following new running times for solving these problems.

**Theorem 1** (Principal Component Projection). *For any* $\epsilon \in (0,1)$, *there is an* $\epsilon$-*approximate PCP algorithm (see Definition 1)* ISPCP$(\mathbf{A}, \mathbf{v}, \lambda, \gamma, \epsilon, \delta)$ *specified in Algorithm 5 with runtime*

$$\tilde{O}\left(\mathrm{nnz}(\mathbf{A}) + \sqrt{\mathrm{nnz}(\mathbf{A}) \cdot d \cdot \mathrm{sr}(\mathbf{A})} \kappa / \gamma\right).$$

**Theorem 2** (Principal Component Regression). *For any* $\epsilon \in (0,1)$, *there is an* $\epsilon$-*approximate PCR algorithm (see Definition 2)* ISPCR$(\mathbf{A}, \mathbf{b}, \lambda, \gamma, \epsilon, \delta)$ *specified in Algorithm 6 with runtime*

$$\tilde{O}\left(\mathrm{nnz}(\mathbf{A}) + \sqrt{\mathrm{nnz}(\mathbf{A}) \cdot d \cdot \mathrm{sr}(\mathbf{A})} \kappa / \gamma\right).$$

We achieve these results by introducing a technique we call *asymmetric SVRG* to solve squared systems $[(\mathbf{A}^\top\mathbf{A} - c\mathbf{I})^2 + \mu^2\mathbf{I}]\mathbf{x} = \mathbf{v}$ with $c \in [0, \lambda_1]$. The resulting algorithm is closely related to the SVRG algorithm for monotone operators in Palaniappan and Bach [13], but involves a more fine-grained error analysis. This analysis coupled with approximate proximal point [14] or Catalyst [15] yields the following result (see Section 3 for more details).

**Theorem 3** (Squared Solver). *For any $\epsilon \in (0,1)$, there is an $\epsilon$-approximate squared ridge regression solver (see Definition 3) using* AsySVRG($\mathbf{M}, \hat{\mathbf{v}}, \mathbf{z}_0, \epsilon \|\mathbf{v}\|, \delta$) *that runs in time*

$$\tilde{O}\left(\text{nnz}(\mathbf{A}) + \sqrt{\text{nnz}(\mathbf{A})d \cdot \text{sr}(\mathbf{A})} \lambda_1 / \mu\right).$$

When the eigenvalues of $\mathbf{A}^\top \mathbf{A} - c\mathbf{I}$ are bounded away from 0, such a solver can be utilized to solve non-PSD linear systems in form $(\mathbf{A}^\top \mathbf{A} - c\mathbf{I})\mathbf{x} = \mathbf{v}$ through preconditioning and considering the corresponding problem $(\mathbf{A}^\top \mathbf{A} - c\mathbf{I})^2 \mathbf{x} = (\mathbf{A}^\top \mathbf{A} - c\mathbf{I})\mathbf{v}$ (see Corollary 1).

**Corollary 1.** *Given $c \in [0, \lambda_1]$, and a non-PSD system $(\mathbf{A}^\top \mathbf{A} - c\mathbf{I})\mathbf{x} = \mathbf{v}$ and an initial point $\mathbf{x}_0$, for arbitrary $c$ satisfying $(\mathbf{A}^\top \mathbf{A} - c\mathbf{I})^2 \succeq \mu^2 \mathbf{I}, \mu > 0$, there is an algorithm returns with probability $1 - \delta$ a solution $\tilde{\mathbf{x}}$ such that $\|\tilde{\mathbf{x}} - (\mathbf{A}^\top \mathbf{A} - c\mathbf{I})^{-1}\mathbf{v}\| \leq \epsilon \|\mathbf{v}\|$, within runtime $\tilde{O}\left(\text{nnz}(\mathbf{A}) + \sqrt{\text{nnz}(\mathbf{A})d \cdot \text{sr}(\mathbf{A})} \lambda_1 / \mu\right).$*

Another byproduct of the rational approximation used in the paper is a nearly-linear runtime for computing an $\epsilon$-approximate square-root of PSD matrix $\mathbf{M} \succeq \mathbf{0}$ applied to an arbitrary vector.

**Theorem 4** (Square-root Computation). *For any $\epsilon \in (0,1)$, given $\mu \mathbf{I} \preceq \mathbf{M} \preceq \lambda \mathbf{I}$, there is an $\epsilon$-approximate square-root solver (see Definition 4)* SquareRoot($\mathbf{M}, \mathbf{v}, \epsilon, \delta$) *that runs in time*

$$\tilde{O}(\text{nnz}(\mathbf{M}) + \mathcal{T})$$

*where $\mathcal{T}$ is the runtime for solving $(\mathbf{M} + \kappa \mathbf{I})\mathbf{x} = \mathbf{v}$ for arbitrary $\mathbf{v} \in \mathbb{R}^n$ and $\kappa \in [\tilde{\Omega}(\mu), \tilde{O}(\lambda)]$.*

### 1.4 Comparison to Previous Work

**PCP and PCR**: The starting point for our paper is the work of [7], which provided the first nearly linear time algorithm for the problem with constant eigengap by reducing the problem to finding the best polynomial approximation to sign function and solving a sequence of regression problems. It was improved by [8] and then [17]. These algorithms were first to achieve input sparsity for eigenspaces of any non-trivial size, but with super-linear running times whenever the eigenvalue-gap is super-constant. Departing from their polynomial approximation, we use rational function as approximant and reduce to different subproblems to get new algorithms with better running time guarantee in some regime. See Table 1 for a comparison between those results and ours.

Table 1: Comparison with previous PCP/PCR runtimes. (Notations same as in Section 1.3.)

| Algorithm | Runtime |
|---|---|
| FMMS16 [7] | $\tilde{O}\left(\frac{1}{\gamma^2}(\text{nnz}(\mathbf{A}) + d \cdot \text{sr}(\mathbf{A})\kappa)\right)$ |
| AL17 [8], MMS18 [17] | $\tilde{O}\left(\frac{1}{\gamma}(\text{nnz}(\mathbf{A}) + d \cdot \text{sr}(\mathbf{A})\kappa)\right)$ |
| Theorems 1 and 2 | $\tilde{O}\left(\text{nnz}(\mathbf{A}) + \sqrt{\text{nnz}(\mathbf{A}) \cdot d \cdot \text{sr}(\mathbf{A})}\kappa/\gamma\right)$ |

**Asymmetric SVRG and Iterative Methods for Solving Linear Systems**: Variance reduction or varianced reduced iterative methods (e.g. SVRG [12] is a powerful tool for improving convergence of stochastic methods. There has been work that used SVRG to develop primal-dual algorithms for solving saddle-point problems and extended it to monotone operators [13]. Our asymmetric SVRG solver can be viewed as an instance of their algorithm. We obtain improved running time analysis by performing a more fine-grained analysis exploiting problem structure. Especially, we provide Section 1.4 to comparing the effectiveness of our asymmetric SVRG solver with some classic optimization methods for solving non-PSD system $(\mathbf{A}^\top \mathbf{A} - c\mathbf{I})\mathbf{x} = \mathbf{v}$ satisfying $(\mathbf{A}^\top \mathbf{A} - c\mathbf{I})^2 \succeq \mu^2 \mathbf{I}, \mu > 0$ (full discussion in Section 3 and Appendix C.4).

Table 2: Comparison for runtimes of solving non-PSD system $(\mathbf{A}^\top \mathbf{A} - c\mathbf{I})\mathbf{x} = \mathbf{v}$.

| Algorithm | Runtime |
|---|---|
| AGD applied to squared counterpart | $\tilde{O}(\text{nnz}(\mathbf{A})\lambda_1 / \mu)$ |
| SVRG applied to squared counterpart | $\tilde{O}(\text{nnz}(\mathbf{A}) + \text{nnz}(\mathbf{A})^{3/4} d^{1/4} \text{sr}(\mathbf{A})^{1/2} \lambda_1 / \mu)$ |
| Asymmetric SVRG (Corollary 1) | $\tilde{O}(\text{nnz}(\mathbf{A}) + \sqrt{\text{nnz}(\mathbf{A})d \cdot \text{sr}(\mathbf{A})} \lambda_1 / \mu)$ |

**Fast Matrix Multiplication**: One can also use fast-matrix multiplication (FMM) to possibly speed up all runtimes for PCA, PCR, and PCP, mainly by computing $\mathbf{A}^\top \mathbf{A}$ in $O(nd^\omega)$ time and SVD of this matrix in an additional $O(d^\omega)$ time [18] where $\omega < 2.379$ [19] is the matrix multiplication constant. Given the well-known practicality concerns of methods using fast matrix multiplication, we focus much of our comparison on methods that do not use FMM.

## 1.5 Paper Organization

The remainder of the paper is organized as follows. In Section 2, we reduce the PCP problem[2] to matrix sign approximation and study the property of Zolotarev rational function used in approximation. In Section 3, we develop the asymmetric and squared linear system solvers using variance reduction and show the theoretical guarantee to prove Theorem 3, and correspondingly Corollary 1. In Section 4, we conduct experiments and compare with previous methods to show efficacy of proposed algorithms. We conclude the paper in Section 5.

## 2 PCP through Matrix Sign Approximation

Here we provide our reductions from PCP to sign function approximation. We consider the rational approximation $r(x)$ found by Zolotarev [10] and study its properties for efficient (Theorem 5) and stable (Lemma 5) algorithm design to reduce the problem to solving squared ridge regressions.

Throughout the section, we denote sign function as $\mathrm{sgn}(x): \mathbb{R} \to \mathbb{R}$, where $\mathrm{sgn}(x) = 1$ whenever $x > 0$, $\mathrm{sgn}(x) = -1$ whenever $x < 0$, and $\mathrm{sgn}(0) = 0$. $\mathcal{P}_k \stackrel{\text{def}}{=} \{a_k x^k + \cdots + a_1 x + a_0 | a_k \neq 0\}$ denote class of degree-$k$ polynomials. $\mathcal{R}_{m,n} \stackrel{\text{def}}{=} \{r_{m,n} | r_{m,n} = p_m/q_n, p_m \in \mathcal{P}_m, q_n \in \mathcal{P}_n\}$ denote class of $(m,n)$-degree (or referred to as $\max\{m,n\}$-degree) rational functions.

For the PCP problem (see Definition 1), we need an efficient algorithm that can approximately apply $\mathbf{P}_\lambda$ to any given vector $\mathbf{v} \in \mathbb{R}^d$. Consider the shifted matrix $\mathbf{A}^\top \mathbf{A} - \lambda \mathbf{I}$ so that its eigenvalues are shifted to $[-1,1]$ with $\lambda$ mapping to 0. Previous work has shown [7, 8] solving PCP can be reduced to finding $f(x)$ that approximates sign function $\mathrm{sgn}(x)$ on $[-1,1]$, formally through the following reduction.

**Lemma 1** (Reduction: from PCP to Matrix Sign Approximation). *Given a function $f(x)$ that $2\epsilon$-approximates $\mathrm{sgn}(x)$:*

$$|f(x) - \mathrm{sgn}(x)| \leq 2\epsilon, \forall |x| \in [\lambda\gamma, 1] \quad and \quad |f(x)| \leq 1, \forall x \in [-1,1], \tag{3}$$

*then $\widetilde{\mathbf{v}} = \frac{1}{2}\left(f(\mathbf{A}^\top \mathbf{A} - \lambda \mathbf{I}) + \mathbf{I}\right)\mathbf{v}$ is an $\epsilon$-approximate PCP solution satisfying (1).*

However, instead of approximating $\mathrm{sgn}(x)$ with polynomials as in previous work [7, 8], where the optimal degree for achieving condition $|f(x) - \mathrm{sgn}(x)| \leq 2\epsilon, \forall |x| \in [\gamma, 1]$ is proved to be $\tilde{O}(1/\gamma)$ in [9], we use Zolotarev rational function for approximation. This brings down the degree to $\tilde{O}(\log(1/\lambda\gamma))$, leading to the nearly input sparsity runtime improvement in the paper.

Formally, Zolotarev rationals are defined as the optimal solution $r_k^\gamma(x) = x \cdot p(x^2)/q(x^2) \in \mathcal{R}_{2k+1, 2k}$ for the optimization problem:

$$\max_{p,q \in P_k} \min_{\gamma \leq x \leq 1} \quad x\frac{p(x^2)}{q(x^2)} \qquad \text{s.t.} \qquad x\frac{p(x^2)}{q(x^2)} \leq 1, \forall x \in [0,1] \tag{4}$$

Zolotarev [10] showed this optimization problem (up to scaling) is equivalent to solving

$$\min_{r \in \mathcal{R}_{2k+1,2k}} \max_{|x| \in [\gamma,1]} |\mathrm{sgn}(x) - r(x)| .$$

Further Zolotarev [11] showed that the analytic formula of $r_k^\gamma$ is given by

$$r_k^\gamma(x) = Cx \prod_{i \in [k]} \frac{x^2 + c_{2i}}{x^2 + c_{2i-1}} \text{ with } c_i \stackrel{\text{def}}{=} \gamma^2 \frac{\mathrm{sn}^2(\frac{iK'}{2k+1}; \gamma')}{\mathrm{cn}^2(\frac{iK'}{2k+1}; \gamma')}, i \in [2k]. \tag{5}$$

and $C$ is the rescaling parameter to make sure $1 - r_k^\gamma(\gamma) = -(1 - r_k^\gamma(1))$. Note all coefficients are dependent of degree $k$ and range $\gamma$. The explicit formulas for $c_i, K', \gamma'$ are shown in Appendix B.1.

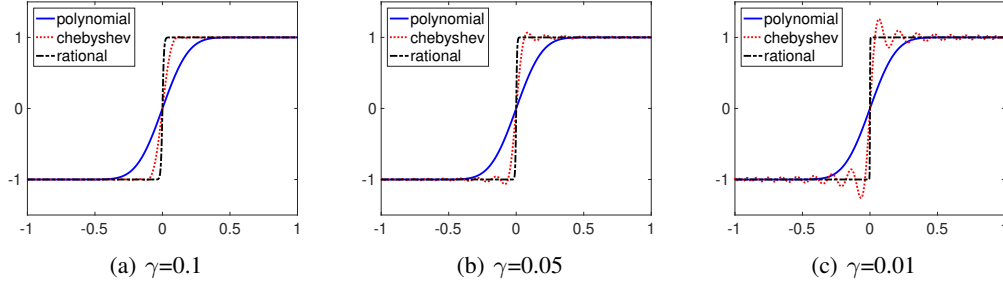

<center>(a) $\gamma=0.1$        (b) $\gamma=0.05$        (c) $\gamma=0.01$</center>

<center>Figure 1: same degree = 21, different $\gamma$</center>

This rational polynomial approximates $\mathrm{sgn}(x)$ on range $|x|\in[\gamma,1]$ with error decaying exponentially with degree, as formally characterized by the following theorem.

**Theorem 5** (Rational Approximation Error). *For any given $\epsilon\in(0,1)$, when $k\geq\Omega(\log(1/\epsilon)\log(1/\gamma))$, it holds that $\max_{|x|\in[\gamma,1]}|\mathrm{sgn}(x)-r_k^\gamma(x)|\leq 2\epsilon$.*

As a quick illustration, Fig. 1 shows a comparison between the approximation errors of Zolotarev rational function, polynomial used in [7] and chebyshev polynomial used in [8] with same degree.

Treating $r_k^{\lambda\gamma}$ with $k\geq\Omega(\log(1/\epsilon)\log(1/\lambda\gamma))$ as the desired $f$ in Lemma 1, it suffices to compute

$$r_k^{\lambda\gamma}((\mathbf{A}^\top\mathbf{A}-\lambda\mathbf{I}))\mathbf{v}=C(\mathbf{A}^\top\mathbf{A}-\lambda\mathbf{I})\prod_{i=1}^{k}\frac{(\mathbf{A}^\top\mathbf{A}-\lambda\mathbf{I})^2+c_{2i}\mathbf{I}}{(\mathbf{A}^\top\mathbf{A}-\lambda\mathbf{I})^2+c_{2i-1}\mathbf{I}}\mathbf{v}.$$

To compute this formula approximately, we need to solve squared linear systems of the form $((\mathbf{A}^\top\mathbf{A}-\lambda\mathbf{I})^2+c_{2i-1}\mathbf{I})\mathbf{x}=\mathbf{v}$, the hardness of which is determined by the size of $c_{2i-1}(>0)$. The larger $c_{2i-1}$ is, the more positive-definite (PD) the system becomes, and the faster we can solve it. The following lemma shows that, the $r_k^{\lambda\gamma}$ we need to use has coefficients $c_i=\tilde{\Omega}(1/\lambda^2\gamma^2)$ when $k=\Theta(\log(1/\epsilon)\log(1/\lambda\gamma))$.

**Lemma 2** (Bounding $c_i$). *For $r_k^{\lambda\gamma}$, coefficients $c_i$ are nondecreasing in $i$, $\forall i\in[2k]$. Also, $\exists$ some constant $0<\beta_2,\beta_3<\infty$, such that $c_1\geq\beta_2\frac{\lambda^2\gamma^2}{k^2}$, $c_{2k}\leq\beta_3 k^2$.*

Given a squared ridge regression solver $\mathtt{RidgeSquare}(\mathbf{A},\lambda,c_{2i-1},\mathbf{v},\epsilon,\delta)$ (See Section 3), we can get an $\epsilon$-approximate PCP algorithm $\mathtt{ISPCP}(\mathbf{A},\mathbf{v},\lambda,\gamma,\epsilon,\delta)$ shown in Algorithm 5 and its theoretical guarantee in Theorem 1. Using the known reduction [7, 8] from PCP to PCR solver, this also gives results in Theorem 2. We refer readers to Appendix D for parameter choice and corresponding proofs.

---

**Algorithm 1:** $\mathtt{ISPCP}(\mathbf{A},\mathbf{v},\lambda,\gamma,\epsilon,\delta)$

---

**Input:** $\mathbf{A}$ data matrix, $\mathbf{v}$ projecting vector, $\lambda$ threshold, $\gamma$ eigengap, $\epsilon$ accuracy, $\delta$ probability.
**Parameter:** degree $k$ (Theorem 5), coefficients $\{c_i\}_{i=1}^{2k}$, $C$ (Eqn. (5)), accuracy $\epsilon_1$ (Appendix D)
**Output:** A vector $\widetilde{\mathbf{v}}$ that solves PCP $\epsilon$-approximately.

1 **for** $i\leftarrow 1$ *to* $k$ **do**
2     $\widetilde{\mathbf{v}}\leftarrow(\mathbf{A}^\top\mathbf{A}-\lambda\mathbf{I})^2\widetilde{\mathbf{v}}+c_{2i}\widetilde{\mathbf{v}}$
3     $\widetilde{\mathbf{v}}\leftarrow\mathtt{RidgeSquare}(\mathbf{A},\lambda,c_{2i-1},\widetilde{\mathbf{v}},\epsilon_1,\delta/k)$
4 $\widetilde{\mathbf{v}}\leftarrow C(\mathbf{A}^\top\mathbf{A}-\lambda\mathbf{I})\widetilde{\mathbf{v}}$
5 $\widetilde{\mathbf{v}}\leftarrow\frac{1}{2}(\mathbf{v}+\widetilde{\mathbf{v}})$

---

## 3 SVRG for Solving Asymmetric / Squared Systems

In this section, we reduce solving squared systems into solving asymmetric systems (Lemma 3) and develop SVRG-type solvers (Algorithm 2) for them. We study its theoretical guarantees in both general (Theorems 6 and 7) and our specific case (Theorem 8). We defer all proofs to Appendix C.

In Section 2, we get low-degree rational function approximation at the cost of more complicated subproblems to solve. Indeed, instead of solving ridge-regression-type subproblems $(\mathbf{A}^\top\mathbf{A}+\lambda\mathbf{I})\mathbf{x}=\mathbf{v}$

<center>6</center>

as in previous work [7, 8], we need to solve squared systems in the following form:

$$[(\mathbf{A}^\top\mathbf{A}-c\mathbf{I})^2+\mu^2\mathbf{I}]\mathbf{x}=\mathbf{v}, \text{ with } \mathbf{A}\in\mathbb{R}^{n\times d}, \mathbf{v}\in\mathbb{R}^d, \mu>0, c\in[0,\lambda_1]. \tag{6}$$

When the squared system is ill-conditioned (i.e. when $\lambda_1/\mu\gg0$), previous state-of-the-art methods can have fairly large running times. As shown in Section 1.4 and proved in Appendix C.4, Accelerated Gradient Descent [20] applied to solving system $((\mathbf{A}^\top\mathbf{A}-c\mathbf{I})^2+\mu^2\mathbf{I})\mathbf{x}=\mathbf{v}$ gives a runtime $\tilde{O}(\mathrm{nnz}(\mathbf{A})\lambda_1/\mu)$, which is not nearly linear in $\mathrm{nnz}(\mathbf{A})$. Applying the standard SVRG [12] technique to the same system leads to a runtime $\tilde{O}(\mathrm{nnz}(\mathbf{A})+d\cdot\mathrm{sr}^2(\mathbf{A})\lambda_1^4/\mu^4)$, where $\mathrm{sr}^2(\mathbf{A})\lambda_1^4/\mu^4$ comes from the high variance in sampling $\mathbf{a}_i\mathbf{a}_i^\top\mathbf{a}_j\mathbf{a}_j^\top$ from $(\mathbf{A}^\top\mathbf{A})^2$ independently.

Thus rather than working with the squared system directly, we propose to consider (equivalently) a larger dimensional space where we develop estimators with lower variance at the cost of asymmetry, formally in the reduction below.

**Lemma 3** (Reducing Squared Systems to Asymmetric Systems). *Define $\mathbf{z}^*$ as the solution to the following asymmetric linear system:*

$$\begin{pmatrix} \mathbf{I} & -\frac{1}{\mu}(\mathbf{A}^\top\mathbf{A}-c\mathbf{I}) \\ \frac{1}{\mu}(\mathbf{A}^\top\mathbf{A}-c\mathbf{I}) & \mathbf{I} \end{pmatrix}\mathbf{z}=\begin{pmatrix} \mathbf{0} \\ \mathbf{v}/\mu^2 \end{pmatrix}. \tag{7}$$

*If we are given a solver that returns with probability $1-\delta$ a solution $\widetilde{\mathbf{z}}$ satisfying $\|\widetilde{\mathbf{z}}-\mathbf{z}^*\|_2\leq\epsilon$ within runtime $\mathcal{T}(\epsilon,\delta)$, then we can use it to get an $\epsilon$-approximate squared ridge regression solver (see Definition 3) with runtime $\mathcal{T}(\epsilon\|\mathbf{v}\|,\delta)$ .*

### 3.1 SVRG for General Asymmetric Linear System Solving

The general goal for this section is to solve the general asymmetric system with PSD symmetric part, formally defined as:

$$\text{solve } \mathbf{Mz}=\hat{\mathbf{v}} \text{ with } \hat{\mathbf{v}}\in\mathbb{R}^a, \mathbf{M}\in\mathbb{R}^{a\times a}, \mathbf{M}=\sum_{i\in[n]}\mathbf{M}_i, \|\mathbf{M}_i\|\leq L_i, \frac{1}{2}(\mathbf{M}^\top+\mathbf{M})\succeq\mu\mathbf{I} \tag{8}$$

For simplicity, we denote $\mathcal{T}_{\mathrm{mv}}(\mathbf{M}_i)$ as the cost of the matrix-vector product of $\mathbf{M}_i\mathbf{x}$ for any $\mathbf{x}$ and $\mathcal{T}=\max_{i\in[n]}\mathcal{T}_{\mathrm{mv}}(\mathbf{M}_i)$. All results in this subsection can be viewed as a variant of Palaniappan and Bach [13] and can be recovered by their slightly different algorithm which used proximal methods.

Using the idea of variance-reduced sampling [12]: At step $t$, we sample $i_t\in[n$ with probability $p_{i_t}=L_{i_t}/(\sum_{i\in[n]}L_i)$ independently and conduct update

$$\mathbf{z}_{t+1}:=\mathbf{z}_t-\frac{\eta}{p_{i_t}}\big(\mathbf{M}_{i_t}\mathbf{z}_t-\mathbf{M}_{i_t}\mathbf{z}_0+p_{i_t}(\mathbf{Mz}_0-\hat{\mathbf{v}})\big). \tag{9}$$

---

**Algorithm 2:** AsySVRG$(\mathbf{M},\hat{\mathbf{v}},\mathbf{z}_0,\epsilon,\delta)$

**Input:** $\mathbf{M}\in\mathbb{R}^{a\times a}$, $\hat{\mathbf{v}}\in\mathbb{R}^a$, $\mathbf{z}_0\in\mathbb{R}^a$, $\epsilon$ desired accuracy, $\delta$ probability parameter.
**Output:** $\mathbf{z}_0^{(Q+1)}\in\mathbb{R}^a$.
1 Set $\eta=\mu/4(\sum_{i\in[n]}L_i)^2$, $T=\lceil(\sum_{i\in[n]}L_i)^2/\mu^2\rceil$, $p_i=L_i/(\sum_{i\in[n]}L_i), i\in[n]$ unless specified
2 **for** $q=1$ *to* $Q=\Theta(\log(1/\epsilon\delta))$ **do**
3    **for** $t\leftarrow1$ *to* $T$ **do**
4       Sample $i_t\sim[n]$ according to $\{p_i\}_{i=1}^n$
5       $\mathbf{z}_t^{(q)}\leftarrow\mathbf{z}_{t-1}^{(q)}-\eta/p_{i_t}(\mathbf{M}_{i_t}\mathbf{z}_{t-1}^{(q)}-\mathbf{M}_{i_t}\mathbf{z}_0^{(q)}+p_{i_t}(\mathbf{Mz}_0^{(q)}-\hat{\mathbf{v}}))$
6    $\mathbf{z}_0^{(q+1)}=\frac{1}{T}\sum_{t=1}^T\mathbf{z}_t^{(q)}$

---

The full pseudocode is shown in Algorithm 2. It has the following theoretical guarantee.

**Theorem 6** (General Asymmetric SVRG Solver). *For asymmetric system $\mathbf{Mz}=\hat{\mathbf{v}}$ in (8), there is a solver* AsySVRG$(\mathbf{M},\hat{\mathbf{v}},\mathbf{z}_0,\epsilon,\delta)$ *as specified in Algorithm 2 that returns with high probability $\geq1-\delta$ a vector $\widetilde{\mathbf{z}}$ such that $\|\widetilde{\mathbf{z}}-\mathbf{M}^{-1}\hat{\mathbf{v}}\|\leq\epsilon$, within runtime $\tilde{O}(\mathrm{nnz}(\mathbf{M})+\mathcal{T}(\sum_{i\in[n]}L_i)^2/\mu^2)$.*

Using approximate proximal point [14] or Catalyst [15], when $\mathrm{nnz}(\mathbf{M})\leq\mathcal{T}(\sum_{i\in[n]}L_i)^2/\mu^2$, we can further improve this running time to the following:

**Theorem 7** (Accelerated Asymmetric SVRG Solver). *Under (8), when* $\mathrm{nnz}(\mathbf{M}) \leq \mathcal{T}(\sum_{i \in [n]} L_i)^2/\mu^2$, *the algorithm can be further accelerated to return with high probability* $\geq 1-\delta$ *an approximate solution* $\widetilde{\mathbf{z}}$ *satisfying* $\|\widetilde{\mathbf{z}} - \mathbf{M}^{-1}\hat{\mathbf{v}}\| \leq \epsilon$, *within runtime* $\tilde{O}\big(\sqrt{\mathrm{nnz}(\mathbf{M})\mathcal{T}}(\sum_{i \in [n]} L_i)/\mu\big)$.

## 3.2 Asymmetric Linear System Solving for Squared System Solver

From Lemma 3, the asymmetric linear system we actually need to solve is $\mathbf{M}\mathbf{z} = \hat{\mathbf{v}}$, where

$$\mathbf{M} = \begin{pmatrix} \mathbf{I} & -\frac{1}{\mu}(\mathbf{A}^\top \mathbf{A} - c\mathbf{I}) \\ \frac{1}{\mu}(\mathbf{A}^\top \mathbf{A} - c\mathbf{I}) & \mathbf{I} \end{pmatrix} \text{ and } \hat{\mathbf{v}} = \big(\mathbf{0}, \mathbf{v}/\mu^2\big)^\top. \tag{10}$$

Through a more fine-grained analysis shown in Appendix C.2, $\texttt{AsySVRG}(\mathbf{M}, \hat{\mathbf{v}}, \mathbf{z}_0, \epsilon, \delta)$ with particular choices of $\mathbf{M}_i, \{p_i\}_{i \in [n]}, \eta, T$ can have a better runtime guarantee and be accelerated using similar idea as in the general case. This is stated formally in the following theorem.

**Theorem 8** (Particular Asymmetric SVRG Solver). *Set* $p_i = \|\mathbf{a}_i\|^2/\|\mathbf{A}\|_F^2$, $\eta = \mu^2/2\lambda_1\|\mathbf{A}\|_F^2$, $T = \lceil 2\|\mathbf{A}\|_F^2 \lambda_1/\mu^2 \rceil$ *and*

$$\mathbf{M}_i := \begin{pmatrix} \frac{\|\mathbf{a}_i\|^2}{\|\mathbf{A}\|_F^2} I & -\frac{1}{\mu}(\mathbf{a}_i \mathbf{a}_i^\top - c\frac{\|\mathbf{a}_i\|^2}{\|\mathbf{A}\|_F^2}\mathbf{I}) \\ \frac{1}{\mu}(\mathbf{a}_i \mathbf{a}_i^\top - c\frac{\|\mathbf{a}_i\|^2}{\|\mathbf{A}\|_F^2})\mathbf{I} & \frac{\|\mathbf{a}_i\|^2}{\|\mathbf{A}\|_F^2}\mathbf{I} \end{pmatrix}, \forall i \in [n].$$

*Then* $\texttt{AsySVRG}(\mathbf{M}, \hat{\mathbf{v}}, \mathbf{z}_0, \epsilon, \delta)$ *as specified in Algorithm 2 returns with probability* $\geq 1 - \delta$ *an* $\epsilon$-*approximate solution* $\widetilde{\mathbf{z}}$ *satisfying* $\|\widetilde{\mathbf{z}} - \mathbf{M}^{-1}\hat{\mathbf{v}}\| \leq \epsilon$ *within runtime* $\tilde{O}\big(\mathrm{nnz}(\mathbf{A}) + d \cdot \mathrm{sr}(\mathbf{A})\lambda_1^2/\mu^2\big)$. *An accelerated variant of it improves the runtime to* $\tilde{O}\big(\lambda_1\sqrt{\mathrm{nnz}(\mathbf{A})d \cdot \mathrm{sr}(\mathbf{A})}/\mu\big)$ *when* $\mathrm{nnz}(\mathbf{A}) \leq d \cdot \mathrm{sr}(\mathbf{A})\lambda_1^2/\mu^2$.

Picking $c = \lambda, \mu^2 = c_{2i-1} = \tilde{\Omega}(1/\lambda^2\gamma^2)$ (see Lemma 5) in (10), we know under minimal trasformations $\texttt{AsySVRG}(\mathbf{M}, \hat{\mathbf{v}}, \mathbf{z}_0, \epsilon\|\mathbf{v}\|, \delta)$ is equivalent to an $\epsilon$-approximate squared solver $\texttt{RidgeSquare}(\mathbf{A}, \lambda, c_{2i-1}, \mathbf{v}, \epsilon, \delta)$, with worst-case runtime $\widetilde{O}\big(\mathrm{nnz}(\mathbf{A}) + d \cdot \mathrm{sr}(\mathbf{A})\frac{\kappa^2}{\gamma^2}\big)$. (See Appendix C.3 and Algorithm 4 for details.)

## 4 Numerical Experiments

We evaluate our proposed algorithms following the settings in Frostig et al. [7], Allen-Zhu and Li [8]. As the runtimes in Theorems 1 and 2 show improvement compared with the ones in previous work [7, 8] when $\mathrm{nnz}(\mathbf{A})/\gamma \gg d^2\kappa^2/\gamma^2$, we pick the data matrix $\mathbf{A}$ such that $\kappa = \Theta(1)$ and $n \gg \frac{d}{\gamma}$ to corroborate the theoretical results.

Since experiments in several papers [7, 8] have studied the reduction from PCR to PCP (see Lemma 13), here we only show results regarding solving PCP problems. In all figures below, the $y$-axis denotes the relative error measured in $\|\mathcal{A}_{\mathrm{PCP}}(\mathbf{v}) - \mathbf{P}_\lambda \mathbf{v}\|/\|\mathbf{P}_\lambda \mathbf{v}\|$ and $x$-axis denotes the total number of vector-vector products to achieve corresponding accuracy.

**Datasets.** Similar to that in previous work [7, 8], we set $\lambda = 0.5, n = 2000, d = 50$ and form a matrix $\mathbf{A} = \mathbf{U}\mathbf{\Lambda}^{1/2}\mathbf{V} \in \mathbb{R}^{2000 \times 50}$. Here, $\mathbf{U}$ and $\mathbf{V}$ are random orthonormal matrices, and $\mathbf{\Sigma}$ contains randomly chosen singular values $\sigma_i = \sqrt{\lambda_i}$. Referring to $[0, \lambda(1-\gamma)] \cup [\lambda(1+\gamma), 1]$ as the *away-from-$\lambda$ region*, and $\lambda(1-\gamma) \cdot [0.9, 1] \cup \lambda(1+\gamma) \cdot [1, 1.1]$ as the *close-to-$\lambda$ region*, we generate $\lambda_i$ differently to simulate the following three different cases:
i. *Eigengap-Uniform Case*: generate all $\lambda_i$ uniformly in the away-from-$\lambda$ region.
ii. *Eigengap-Skewed Case*: generate half the $\lambda_i$ uniformly in the away-from-$\lambda$ and half uniformly in the close-to-$\lambda$ regions.
iii. *No-Eigengap-Skewed Case*: uniformly generate half in $[0,1]$, and half in the close-to-$\lambda$ region.

**Algorithms.** We implemented the following algorithms and compared them in the above settings:
1. *polynomial*: the $\texttt{PC-Proj}$ algorithm in Frostig et al. [7].
2. *chebyshev*: the $\texttt{QuickPCP}$ algorithm in Allen-Zhu and Li [8].
3. *lanczos*: the algorithm using Lanczos method discussed in Section 8.1 of Musco et al. [17].
4. *rational*: the $\texttt{ISPCP}$ algorithm (see Algorithm 5) proposed in our paper.
5. *rlanczos*: the algorithm using rational Lanczos method [21] combined with ISPCP. (See Appendix E.1 for a more detailed discussion.)

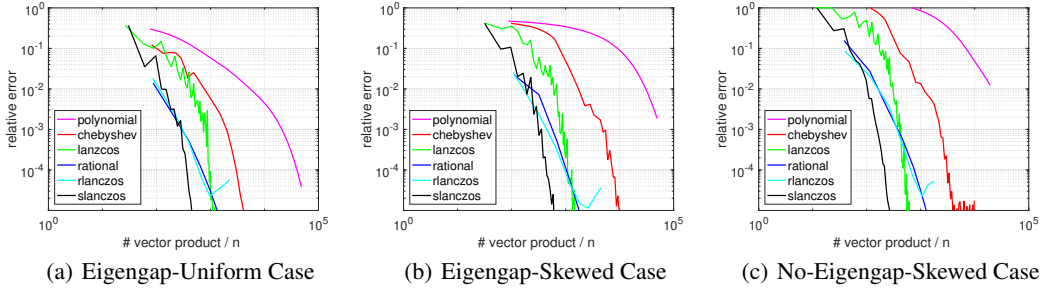

Figure 2: Synthetic Data: $n=2000, d=50, \lambda=0.5, \gamma=0.05$.

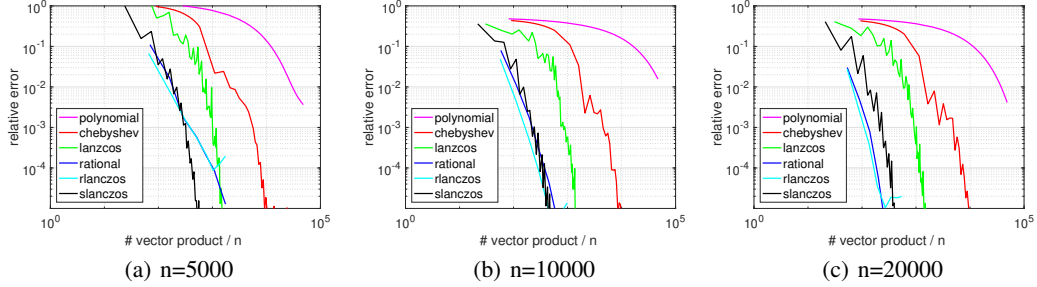

Figure 3: Synthetic Data: Changing $n$, $d=50, \lambda=0.5, \gamma=0.05$. No-Eigengap-Skewed Case.

6. *slanczos*: the algorithm using Lanczos method [17] with changed search space from form $f\left(\frac{x-\lambda}{x+\lambda}\right)$ into $f\left(\frac{(x-\lambda)(x+\lambda)}{(x-\lambda)^2+\gamma(x+\lambda)^2}\right)$ for approximation ($f$ polynomial, $x \leftarrow \mathbf{A}^\top \mathbf{A}$).

We remark that 1-3 are algorithms in previous work; 4 is an exact implementation of ISPCP proposed in the paper; 5, 6 are variants of ISPCP combined with Lanczos method, both using the squared system solver. Algorithms 5, 6 are explained in greater detail in Appendix E.

There are several observations from the experiments:
• For different eigenvalue distributions, (4-6) in general outperform all existing methods (1-3) in most accuracy regime in terms of number of vector products as shown in Fig. 2.
• In no-eigengap case, all methods get affected in precision. This is due to the projection error of eigenvalues very close to eigengap, which simple don't exist in Eigengap cases. Nevertheless, (6) is still the most accurate one with least computation cost, as shown in Fig. 2.
• When $n$ gets larger, (4,5) tends to enjoy similar performance, outperforming all other methods including (6), as shown in Fig. 3. This aligns with theory that runtime of (4,5) is dominated by nnz($\mathbf{A}$) while runtime of (6) is dominated by nnz($\mathbf{A}$)$/\sqrt{\gamma}$ (see Theorem 12 for theoretical analysis of *slanczos*), demonstrating the power of nearly-linear runtime of ISPCP proposed.

## 5   Conclusion

In this paper we provided a new linear algebraic primitive, asymmetric SVRG for solving squared ridge regression problems, and showed that it lead to nearly-linear-time algorithms for PCP and PCR. Beyond the direct running time improvements, this work shows that running time improvements can be achieved for fundamental linear-algebraic problems by leveraging stronger subroutines than standard ridge regression. The improvements we obtain for PCP, demonstrated theoretically and empirically, we hope are just the first instance of a more general approach for improving the running time for solving large-scale machine learning problems.

**Acknowledgements**   This research was partially supported by NSF CAREER Award CCF-1844855 and Stanford Graduate Fellowship. We would also like to thank the anonymous reviewers who helped improve the completeness and readability of this paper by providing many helpful comments.

## Footnotes

[1] This can be achieved by getting a constant approximating overestimate $\tilde{\lambda}_1$ of $\mathbf{A}^\top\mathbf{A}$'s top eigenvector $\lambda_1$ through power method in $\tilde{O}(\mathrm{nnz}(\mathbf{A}))$ time, and consider $\mathbf{A} \leftarrow \mathbf{A}/\sqrt{\tilde{\lambda}_1}, \lambda \leftarrow \lambda/\tilde{\lambda}_1, \mathbf{b} \leftarrow \mathbf{b}/\sqrt{\tilde{\lambda}_1}$ instead.

[2]We refer reader to Appendix D.2 for the known reduction from PCR to PCP.

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
