[Supplementary Material · supp_2126.pdf]

## Appendix

## A    Ridge Regression Solver `RidgeReg`

In the reduction from PCP to PCR stated in Section 2, a blackbox solver `RidgeReg(A,`$\lambda$`,s)` is needed as the case of Frostig et al. [7], Allen-Zhu and Li [8]. We formally define it as follows.

**Definition 5** (Ridge Regression Solver). *Given* $\mathbf{s} \in \mathbb{R}^d$, *and exact solution* $\mathbf{x}^* = (\mathbf{A}^\top \mathbf{A} + \mu \mathbf{I})^{-1} \mathbf{s}$, *an algorithm* `RidgeReg(A,`$\mu$`,s)` *is an $\epsilon$-approximate ridge regression blackbox solver if for all $\epsilon > 0$, it returns a solution $\tilde{\mathbf{x}}$ satisfying $\|\tilde{\mathbf{x}} - \mathbf{x}^*\| \le \epsilon \|\mathbf{s}\|$.*

**Theorem 9** (Blackbox Solver for Ridge Regression). *There is a blackbox solver for ridge regression with runtime $\tilde{O}(\mathrm{nnz}(\mathbf{A}) + \sqrt{\mathrm{nnz}(\mathbf{A})d \cdot \mathrm{sr}(\mathbf{A})\kappa_\mu})$.*

An algorithm achieving the aboveboard theoretical guarantee can be found in Shalev-Shwartz and Zhang [22], Frostig et al. [14], Lin et al. [15].

We remark that when depending on structural properties of $\mathbf{A}$ (denoting the maximum row sparsity of $\mathbf{A}$ as $s(\mathbf{A})$ such that $s(\mathbf{A}) \le d$), the running time for ridge regression can be further improved into $\tilde{O}((n + \sqrt{n \cdot \mathrm{sr}(\mathbf{A})\kappa_\mu}) \cdot s(\mathbf{A}))$ through direct acceleration Katyusha [23] or accelerated coordinate descent methods [24]. Using a leverage-score sampling idea [25], one can obtain an even more fine-grained running time $\tilde{O}((n + \sqrt{d \cdot \mathrm{sr}(\mathbf{A})\kappa_\mu}) \cdot s(\mathbf{A}))$ that improves the previous in most settings. Because these algorithms have running time highly depend on structure of $\mathbf{A}$, we didn't use them for computing runtime when treating ridge regression solver as a blackbox.

## B    Proofs for Results in Section 2

*Proof of Lemma 1.* Note that $\mathbf{A}^\top \mathbf{A} = \mathbf{V}\mathbf{\Lambda}\mathbf{V}^\top$ where $\mathbf{\Lambda} = \mathbf{diag}(\lambda_1, \cdots, \lambda_d)$ and each column of $\mathbf{V}$ is $\boldsymbol{\nu}_i, i \in [d]$. We can write $\mathbf{v} = \sum_{i \in [d]} \alpha_i \boldsymbol{\nu}_i$, and therefore $\mathbf{P}_\lambda \mathbf{v} = \sum_{i \in [k]} \alpha_i \boldsymbol{\nu}_i$. This also implies $f(\mathbf{A}^\top \mathbf{A} - \lambda \mathbf{I}) = \mathbf{V} f(\mathbf{\Lambda} - \lambda \mathbf{I})\mathbf{V}^\top$ where $f(\mathbf{\Lambda} - \lambda \mathbf{I}) \overset{\text{def}}{=} \mathbf{diag}(f(\lambda_1 - \lambda), \cdots, f(\lambda_d - \lambda))$.

Now we define $k_1, k_2 \in [n]$ to divide the eigenvalues $\lambda_i$ and corresponding $\boldsymbol{\nu}_i$ into three settings, (1) $\lambda_i \ge \lambda(1 + \gamma)$ for all $i \le k_1$, (2) $\lambda_i \in (\lambda(1 - \gamma), \lambda(1 + \gamma))$ for all $k_1 < i \le k_2$, and (3) $\lambda_i \le \lambda(1 - \gamma)$ for all $i > k_2$. Since by assumption $0 < \lambda < \lambda_1 \in [1/2, 1]$ and $\gamma \in (0,1)$, it holds that $\lambda_i - \lambda \in [\lambda\gamma, 1]$ when $i \le k_1$). Similarly, $\lambda_i - \lambda \in [-1, -\lambda\gamma]$ when $i < k_2$ and $\lambda_i - \lambda \in (-\lambda\gamma, \lambda\gamma)$ when $k_1 < i \le k_2$. Consequently, $|f(\lambda_i - \lambda) - \mathrm{sgn}(\lambda_i - \lambda)| \le 2\epsilon, \forall \lambda_i \notin (\lambda(1 - \gamma), \lambda(1 + \gamma))$ and we have that $|f(\lambda_i - \lambda) - \mathrm{sgn}(\lambda_i - \lambda)| \le 2, \forall \lambda_i \in (\lambda(1 - \gamma), \lambda(1 + \gamma))$.

Noticing $\widetilde{\mathbf{v}} = \frac{1}{2}\mathbf{V}(f(\mathbf{\Lambda} - \lambda \mathbf{I}) + \mathbf{I})\mathbf{V}^\top \mathbf{v}$, and thus we have

$$\widetilde{\mathbf{v}} - \mathbf{v} = \frac{1}{2}\mathbf{V}(f(\mathbf{\Lambda} - \lambda \mathbf{I}) + \mathbf{I})\mathbf{V}^\top \mathbf{v} - \mathbf{V}\mathbf{V}^\top \mathbf{v} = \frac{1}{2}\mathbf{V}(f(\mathbf{\Lambda} - \lambda \mathbf{I}) - \mathbf{I})\mathbf{V}^\top \mathbf{v}$$

The result then follows by the following, which shows that all conditions in (1) are satisfied

$$\|\mathbf{P}_{(1+\gamma)\lambda}(\widetilde{\mathbf{v}} - \mathbf{v})\| \le \|\epsilon \mathbf{V}[\alpha_1, \cdots, \alpha_{k_1}, 0, \cdots, 0]^\top\| \le \epsilon \|\mathbf{v}\|,$$

$$\|(\mathbf{I} - \mathbf{P}_{(1-\gamma)\lambda})\tilde{\mathbf{v}}\| \le \|\epsilon \mathbf{V}[0, \cdots, 0, \alpha_{k_2+1}, \cdots, \alpha_d]^\top\| \le \epsilon \|\mathbf{v}\|,$$

$$\|(\mathbf{P}_{(1+\gamma)} - \mathbf{P}_{(1-\gamma)\lambda})(\widetilde{\mathbf{v}} - \mathbf{v})\| \le \|\mathbf{V}[\cdots, 0, \alpha_{k_1+1}, \cdots, \alpha_{k_2}, 0, \cdots]^\top\| = \|(\mathbf{P}_{(1+\gamma)} - \mathbf{P}_{(1-\gamma)\lambda})\mathbf{v}\|.$$

$\square$

### B.1    Analytic Expression of Zolotarev Rational

Here, we give a more detailed discussion of the explicit expression of Zolotarev rational function $r_k^\gamma$ stated in (5). The analytic formula of $r_k^\gamma$ is as follows [10]:

$$r_k^\gamma(x) = Cx \prod_{i=1}^{k} \frac{x^2 + c_{2i}}{x^2 + c_{2i-1}}$$

Here, all the constants depend on the explicit range $|x| \in [\gamma, 1]$ we want to approximate uniformly. $C$ is computed through solving $1 - r_k^\gamma(\gamma) = -(1 - r_k^\gamma(1))$, and coefficients $\{c_i\}_{i=1}^{2k}$ are computed

through Jacobi elliptic coefficients, all as follows:

$$\text{coefficients}\begin{cases} C & \overset{\text{def}}{=} \dfrac{2}{(\gamma\prod_{i\in[k]}\frac{\gamma^2+c_{2i}}{\gamma^2+c_{2i-1}})+(\prod_{i\in[k]}\frac{1+c_{2i}}{1+c_{2i-1}})}, \\[4mm] c_i & \overset{\text{def}}{=} \gamma^2\dfrac{\text{sn}^2(\frac{iK'}{2k+1};\gamma')}{\text{cn}^2(\frac{iK'}{2k+1};\gamma')},\forall i\in\{1,2,\cdots,2k\}. \end{cases}$$

$$\text{numerical constants}\begin{cases} \gamma' & \overset{\text{def}}{=} \sqrt{1-\gamma^2}, \\[2mm] K' & \overset{\text{def}}{=} \int_0^{\pi/2}\dfrac{d\theta}{\sqrt{1-\gamma'^2\sin^2\theta}}, \\[2mm] u & \overset{\text{def}}{=} F(\phi;\gamma')\overset{\text{def}}{=}\int_0^{\phi}\dfrac{d\theta}{\sqrt{1-\gamma'^2\sin^2\theta}}, \\[2mm] \text{sn}(u;\gamma') & \overset{\text{def}}{=} \sin(F^{-1}(u;\gamma'));\ \text{cn}(u;\gamma')\overset{\text{def}}{=}\cos(F^{-1}(u;\gamma')). \end{cases}$$

## B.2 Properties of Zolotarev Rational

To prove Theorem 5, we first state the following classic result about the approximation error of $r_k^{\gamma}$ in Lemma 4.

**Lemma 4** (Approximation Error). *The approximation error of $r_k^{\gamma}$ satisfies:*

$$\max_{|x|\in[\gamma,1]}|\text{sgn}(x)-r_k^{\gamma}(x)|=C_k\rho^{-2k+1} \text{ for some } C_k\in\left[\frac{2}{1+\rho^{-(2k+1)}},\frac{2}{1-\rho^{-(2k+1)}}\right]\subseteq[2,\frac{2}{1-\rho^{-1}}].$$

*where* $\rho\overset{\text{def}}{=}\exp(\frac{\pi K(\mu')}{4K(\mu)})>1, K(\mu)\overset{\text{def}}{=}\int_0^1\frac{dt}{\sqrt{(1-t^2)(1-\mu^2t^2)}}, \mu\overset{\text{def}}{=}\frac{1-\sqrt{\gamma}}{1+\sqrt{\gamma}},\mu'\overset{\text{def}}{=}\sqrt{1-\mu^2}.$

This lemma is simply a restatement of equation (32) in Gonchar [26]. We refer interested readers to detailed derivation and proof there.

It is also crucial to our derivation of Theorem 5 and Lemma 2 that we bound the coefficients in $r_k^{\gamma}$ and $\rho$ in Lemma 4. The following lemma provides the key bounds we use for this purpose.

**Lemma 5** (Bounding Coefficients). *The coefficients defined above have the following order / bounds (all constant are independent of $\lambda,\gamma,k$ and any other problem parameters):*
*(1) There exists constant $0<\beta_1<\infty$, such that $K(\mu)\leq\beta_1(\log(1/\gamma)+1)$,*
*(2) Coefficients $c_i$ are nondecreasing in $i$, $\forall i\in[2k]$. Also, there exists some constants $\beta_2>0,\beta_3<\infty$, such that $c_1\geq\beta_2\frac{\gamma^2}{k^2}$, $c_{2k}\leq\beta_3 k^2,\forall i\in[2k]$.*

*Proof of Lemma 5.* (1) The elliptic integral has taylor series [27] as follows:

$$K(\mu)\overset{\text{def}}{=}\int_0^1\frac{dt}{\sqrt{(1-t^2)(1-\mu^2t^2)}}=\frac{\pi}{2}\sum_{n=0}^{\infty}\left(\frac{(2n)!}{2^{2n}(n!)^2}\right)^2\mu^{2n}$$

Using the Stirling formula $n!\approx\sqrt{2\pi n}(n/e)^n$, $\exists$ constants $C_1,C_2<\infty$ such that,

$$K(\mu)=\frac{\pi}{2}\sum_{n=0}^{\infty}\left(\frac{(2n)!}{2^{2n}(n!)^2}\right)^2\mu^{2n}\leq C_1\sum_{n=1}^{\infty}\frac{\mu^{2n}}{n}$$

$$\leq C_2\int_1^{\infty}\frac{\mu^{2t}}{t}dt\overset{(i)}{\leq}C_2\int_{\sqrt{\gamma}}^{\infty}\frac{e^{-t}}{t}dt$$

$$\overset{(ii)}{=}C_2 E_1(\sqrt{\gamma})$$

where we use $(i)$ $\mu^2\leq(1-\sqrt{\gamma})^2\leq 1-\sqrt{\gamma}\leq\exp(-\sqrt{\gamma})$, and $(ii)$ change of variable $t\leftarrow\sqrt{\gamma}t$ and definition of exponential integral that $E_1(z)\overset{\text{def}}{=}\int_z^{\infty}\frac{e^{-t}}{t}dt$

By the convergence series of exponential integral [28], this can be written for $z>0$ as

$$E_1(z)=C_3-\log(z)-\sum_{k=1}^{\infty}\frac{(-z)^k}{k\cdot k!}$$

where $C_3$ finite is the Euler-Mascheroni constant. Using this, we have

$$K(\mu)\leq C_2 E_1(\sqrt{\gamma})\lesssim\log(1/\gamma), \text{ as } \gamma\to 0,$$

where $\lesssim$ is hiding constant $C$ multiplicatively and this yields (1).

(2) By definition of elliptic integral we have,

$$K' \stackrel{\text{def}}{=} \int_0^{\pi/2} \frac{d\theta}{\sqrt{1-\gamma'^2\sin^2\theta}} \quad \text{and} \quad c_i \stackrel{\text{def}}{=} \gamma^2 \frac{\text{sn}^2(\frac{iK'}{2k+1};\gamma')}{\text{cn}^2(\frac{iK'}{2k+1};\gamma')} \quad \text{for all } i \in [2k].$$

Notice by equivalent definition for each $i \in [2k]$ we have,

$$\frac{\text{sn}(\frac{iK'}{2k+1};\gamma')}{\text{cn}(\frac{iK'}{2k+1};\gamma')} = \tan\phi \text{ where } \frac{iK'}{2k+1} = \int_0^\phi \frac{d\theta}{\sqrt{1-\gamma'^2\sin^2\theta}}.$$

Consequently, we know $c_i$ is monotonously decreasing in $i$ as $\phi$ itself is monotonously decreasing.

Also, since $\gamma'^2 = 1-\gamma^2$, we have $\sqrt{1-\gamma'^2\sin^2\theta} = \sqrt{\cos^2\theta + \gamma^2\sin^2\theta}$ which $\to 1$ when $\theta \to 0$ and $\to \gamma$ when $\theta \to \pi/2$. From that we know

$$\frac{1}{2k+1}\int_0^{\pi/2} \frac{d\theta}{\sqrt{1-\gamma'^2\sin^2\theta}} \leq \int_0^\phi \frac{d\theta}{\sqrt{1-\gamma'^2\sin^2\theta}} \leq \frac{2k}{2k+1}\int_0^{\pi/2} \frac{d\theta}{\sqrt{1-\gamma'^2\sin^2\theta}},$$

and thus

$$\frac{1}{2k+1} \cdot \frac{\pi}{2} \leq \phi \leq \frac{\pi}{2} - \frac{\gamma K'}{2k+1}, \quad \text{where by definition } \gamma K' \in (0, \pi/2).$$

Using monotonicity for $\tan(\phi)$ on $\phi \in [0, \pi/2]$, $\tan(\phi) \geq \phi$ and $\sin(\phi) \geq \phi/2$ for all $\phi \in (0, \pi/2)$, we know $\exists \beta_2, \beta_3 \in (0, \infty)$ such that

$$\frac{\text{sn}^2(\frac{K'}{2k+1};\gamma')}{\text{cn}^2(\frac{K'}{2k+1};\gamma')} \geq \beta_2'/(2k+1)^2 \geq \beta_2/k^2$$

$$\frac{\text{sn}^2(\frac{2kK'}{2k+1};\gamma')}{\text{cn}^2(\frac{2kK'}{2k+1};\gamma')} \leq \beta_3'\left(\frac{2k+1}{\gamma K'}\right)^2 \leq \frac{\beta_3 k^2}{\gamma^2}.$$

By definition of $K'$ we know $K' \geq \Theta(1)$, yielding that $c_1 \geq \beta_2 \gamma^2/k^2$ and $c_{2k} \leq \beta_3 k^2$. $\qquad\square$

Now we use these results to prove Lemma 2 and Theorem 5.

*Proof of Lemma 2.* This is a restatement of (2) in Lemma 5. $\qquad\square$

*Proof of Theorem 5.* We apply Lemma 4 and simply need to show that $C_k\rho^{-2k+1} \leq 2\epsilon$. Since $\mu' \in [0,1]$ we have $K(\mu') \geq \int_0^1 (1-t^2)^{-1/2}dt = \pi/2$. Therefore, by (1) in Lemma 5 we have that for some constant $\beta > 0$ we have $\rho \geq \exp(\beta/(\log(1/\gamma)+1))$ and Lemma 4 then yields that

$$C_k\rho^{-2k+1} \leq \frac{2}{\rho^{2k+1}-1} \leq \frac{2}{\exp(\beta(2k+1))/(\log(1/\gamma)+1))-1}.$$

The result follows as $k \geq \Omega(\log(1/\epsilon)\log(1/\gamma))$. $\qquad\square$

## C Proofs for Results in Section 3

*Proof of Lemma 3.* First we show that $\mathbf{M}$ is invertible. Otherwise, if $\mathbf{z} \neq \mathbf{0}$ lies in the nullspace of $\mathbf{M}$ then $\mathbf{z}^\top(\mathbf{M}^\top + \mathbf{M})\mathbf{z} = 0$. By definition of $\mathbf{M}$ we also have $\mathbf{z}^\top(\mathbf{M}^\top + \mathbf{M})\mathbf{z} \geq 2\mu\|\mathbf{z}\|^2 > 0$, given $\mu > 0$. This leads to a contradiction.

Consequently, letting $\hat{\mathbf{v}} \stackrel{\text{def}}{=} [\mathbf{0}; \mathbf{v}/\mu^2]$ we have that $\mathbf{z}^* = \mathbf{M}^{-1}\hat{\mathbf{v}}$ and therefore

$$\begin{pmatrix} \mathbf{I} & -\frac{1}{\mu}(\mathbf{A}^\top\mathbf{A}-c\mathbf{I}) \\ \mathbf{0} & \mathbf{I} \end{pmatrix}\mathbf{z}^* = \begin{pmatrix} \mathbf{I}^{-1} & \mathbf{0} \\ \mathbf{0} & [\mathbf{I}+\frac{1}{\mu^2}(\mathbf{A}^\top\mathbf{A}-c\mathbf{I})^2]^{-1} \end{pmatrix}\begin{pmatrix} \mathbf{I} & \mathbf{0} \\ -\frac{1}{\mu}(\mathbf{A}^\top\mathbf{A}-c\mathbf{I}) & \mathbf{I} \end{pmatrix}\hat{\mathbf{v}}$$

$$= \begin{pmatrix} \mathbf{0} \\ [\mathbf{I}+\frac{1}{\mu^2}(\mathbf{A}^\top\mathbf{A}-c\mathbf{I})^2]^{-1}\mathbf{v}/\mu^2 \end{pmatrix}.$$

Taking the second half of the equations and write $\mathbf{z}^* = [\mathbf{x}^*; \mathbf{y}^*]$ gives

$$\mathbf{y}^* = \left[ (\mathbf{A}^\top \mathbf{A} - c\mathbf{I})^2 + \mu^2 \mathbf{I} \right]^{-1} \mathbf{v}.$$

This is to say optimal solution $\mathbf{y}$ satisfies $\left( (\mathbf{A}^\top \mathbf{A} - c\mathbf{I})^2 + \mu^2 \mathbf{I} \right) \mathbf{y} = \mathbf{v}$.

As a result, if we have a solver that with high probability $\geq 1 - \delta$ in time $\mathcal{T}(\epsilon', \delta)$ gives an $\epsilon'$-approximate solution $\widetilde{\mathbf{z}} = [\widetilde{\mathbf{x}}, \widetilde{\mathbf{y}}]$ of system $\mathbf{M}\mathbf{z} = \hat{\mathbf{v}}$ measured in $L_2$ norm, i.e. $\|\widetilde{\mathbf{x}} - \mathbf{x}^*\|_2^2 + \|\widetilde{\mathbf{y}} - \mathbf{y}^*\|_2^2 \leq \epsilon'^2$, then as long as $\epsilon'^2 \leq \epsilon^2 \|\mathbf{v}\|^2$, we'll get

$$\|\widetilde{\mathbf{y}} - \mathbf{y}^*\|^2 \leq \epsilon^2 \|\mathbf{v}\|^2,$$

giving an $\epsilon$-approximation solution of squared ridge regression $\left( (\mathbf{A}^\top \mathbf{A} - c\mathbf{I})^2 + \mu^2 \mathbf{I} \right) \mathbf{y} = \mathbf{v}$ in time $\mathcal{T}(\epsilon \|\mathbf{v}\|, \delta)$ with probability $1 - \delta$. □

## C.1 Proofs for Results in Section 3.1

Using the idea of variance-reduced sampling [12]: At step $t$, we sample $i_t \in [n$ with probability $p_{i_t} = L_{i_t} / (\sum_{i \in [n]} L_i)$ independently and conduct update

$$\mathbf{z}_{t+1} := \mathbf{z}_t - \frac{\eta}{p_{i_t}} \left( \mathbf{M}_{i_t} \mathbf{z}_t - \mathbf{M}_{i_t} \mathbf{z}_0 + p_{i_t} (\mathbf{M}\mathbf{z}_0 - \hat{\mathbf{v}}) \right). \tag{11}$$

For the above update, when variance is bounded in a desirable way, we can show the following expected convergence guarantee starting at a given initial point $\mathbf{z}_0$:

**Lemma 6** (Progress per epoch). *When the following variance bound on sampling holds*

$$\sum_{i \in [n]} \frac{1}{p_i} \|\mathbf{M}_i \mathbf{z}_t - \mathbf{M}_i \mathbf{z}_0 + p_i \cdot (\mathbf{M}\mathbf{z}_0 - \hat{\mathbf{v}})\|^2 \leq 2S^2 \left[ \|\mathbf{z}_t - \mathbf{z}^*\|^2 + \|\mathbf{z}_0 - \mathbf{z}^*\|^2 \right],$$

*the sampling method* (11) *with a fixed step-size $\eta$ gives, after $T$ iterations, gets*

$$\mathbb{E} \left\| \frac{1}{T} \sum_{t=0}^{T-1} \mathbf{z}_t - \mathbf{z}^* \right\|^2 \leq \frac{\frac{1}{2T} + \eta^2 S^2}{\eta\mu - \eta^2 S^2} \|\mathbf{z}_0 - \mathbf{z}^*\|^2.$$

*Proof.* For a single step at $t$ as in (11), denote the index we drew as $i_t \in [n]$,

$$\|\mathbf{z}_{t+1} - \mathbf{z}^*\|^2 = \|\mathbf{z}_t - \mathbf{z}^*\|^2 - 2\frac{\eta}{p_{i_t}} \langle \mathbf{M}_{i_t} \mathbf{z}_t - \mathbf{M}_{i_t} \mathbf{z}_0 + p_{i_t} (\mathbf{M}\mathbf{z}_0 - \hat{\mathbf{v}}), \mathbf{z}_t - \mathbf{z}^* \rangle$$

$$+ \frac{\eta^2}{p_{i_t}^2} \|\mathbf{M}_{i_t} \mathbf{z}_t - \mathbf{M}_{i_t} \mathbf{z}_0 + p_{i_t} (\mathbf{M}\mathbf{z}_0 - \hat{\mathbf{v}})\|^2.$$

Taking expectation w.r.t $i_t$ we sample we get,

$$\mathbb{E}_{i_t} \|\mathbf{z}_{t+1} - \mathbf{z}^*\|^2 = \|\mathbf{z}_t - \mathbf{z}^*\|^2 - 2\eta \langle \mathbf{M}\mathbf{z}_t - \hat{\mathbf{v}}, \mathbf{z}_t - \mathbf{z}^* \rangle + \eta^2 \sum_{i \in [n]} \frac{1}{p_i} \|\mathbf{M}_i \mathbf{z}_t - \mathbf{M}_i \mathbf{z}_0 + p_i (\mathbf{M}\mathbf{z}_0 - \hat{\mathbf{v}})\|^2. \tag{12}$$

We bound the second and third terms on RHS respectively. For the second term we use the fact that $(\mathbf{M} + \mathbf{M}^\top)/2 \succeq \mu\mathbf{I}$ by assumption and get

$$-\langle \mathbf{M}\mathbf{z}_t - \hat{\mathbf{v}}, \mathbf{z}_t - \mathbf{z}^* \rangle = -\langle \mathbf{M}(\mathbf{z}_t - \mathbf{z}^*), \mathbf{z}_t - \mathbf{z}^* \rangle \leq -\mu \|\mathbf{z}_t - \mathbf{z}^*\|^2. \tag{13}$$

For the third term using condition of bounded variance

$$\sum_{i \in [n]} \frac{1}{p_i} \|\mathbf{M}_i \mathbf{z}_t - \mathbf{M}_i \mathbf{z}_0 + p_i \cdot (\mathbf{M}\mathbf{z}_0 - \hat{\mathbf{v}})\|^2 \leq 2S^2 \left[ \|\mathbf{z}_t - \mathbf{z}^*\|^2 + \|\mathbf{z}_0 - \mathbf{z}^*\|^2 \right]. \tag{14}$$

Combining (12), (13) and (14) we get,

$$\mathbb{E}_{i_t} \|\mathbf{z}_{t+1} - \mathbf{z}^*\|^2 = \|\mathbf{z}_t - \mathbf{z}^*\|^2 - 2\eta\mu \|\mathbf{z}_t - \mathbf{z}^*\|^2 + 2\eta^2 S^2 [\|\mathbf{z}_t - \mathbf{z}^*\|^2 + \|\mathbf{z}_0 - \mathbf{z}^*\|^2]$$

and equivalently,

$$(2\eta\mu-2\eta^2 S^2)\|\mathbf{z}_t-\mathbf{z}^*\|^2\leq\|\mathbf{z}_t-\mathbf{z}^*\|^2-\mathbb{E}_{i_t}\|\mathbf{z}_{t+1}-\mathbf{z}^*\|^2+2\eta^2 S^2\|\mathbf{z}_0-\mathbf{z}^*\|^2.$$

Taking expectation of $i_{t-1},\cdots,i_0$ respectively, averaging over $t=0,1,\cdots,T-1$ thus telescoping the first terms on RHS, and then rearranging terms, we get

$$\mathbb{E}\left\|\frac{1}{T}\sum_{t=0}^{T-1}\mathbf{z}_t-\mathbf{z}^*\right\|^2\leq\mathbb{E}\left[\frac{1}{T}\sum_{t=0}^{T-1}\|\mathbf{z}_t-\mathbf{z}^*\|^2\right]\leq\frac{\frac{1}{2T}+\eta^2 S^2}{\eta\mu-\eta^2 S^2}\|\mathbf{z}_0-\mathbf{z}^*\|^2.$$

$\square$

In the general setting, the variance bound $S$ satisfies the following.

**Lemma 7** (Variance bound for general case). *When $\|\mathbf{M}_i\|\leq L_i,\forall i\in[n]$, the variance bound* (11) *holds with $S=\sum_{i\in[n]}L_i$, i.e.*

$$\sum_{i\in[n]}\frac{1}{p_i}\|\mathbf{M}_i\mathbf{z}_t-\mathbf{M}_i\mathbf{z}_0+p_i\cdot(\mathbf{M}\mathbf{z}_0-\hat{\mathbf{v}})\|^2\leq 2\left(\sum_{i\in[n]}L_i\right)^2\left[\|\mathbf{z}_t-\mathbf{z}^*\|^2+\|\mathbf{z}_t-\mathbf{z}^*\|^2\right].$$

*Proof.* Note that

$$\sum_{i\in[n]}\frac{1}{p_i}\|\mathbf{M}_i\mathbf{z}_t-\mathbf{M}_i\mathbf{z}_0+p_i\cdot(\mathbf{M}\mathbf{z}_0-\hat{\mathbf{v}})\|^2$$

$$\stackrel{(i)}{=}\sum_{i\in[n]}\left(\frac{1}{p_i}\|(\mathbf{M}_i\mathbf{z}_t-\mathbf{M}_i\mathbf{z}^*)+(\mathbf{M}_i\mathbf{z}^*-\mathbf{M}_i\mathbf{z}_0)+p_i\cdot(\mathbf{M}\mathbf{z}_0-\mathbf{M}\mathbf{z}^*)\|^2\right)$$

$$\stackrel{(ii)}{\leq}\sum_{i\in[n]}\left(\frac{2}{p_i}\|\mathbf{M}_i(\mathbf{z}_t-\mathbf{z}^*)\|^2+2p_i\|\frac{1}{p_i}\mathbf{M}_i(\mathbf{z}^*-\mathbf{z}_0)+\mathbf{M}(\mathbf{z}_0-\mathbf{z}^*)\|^2\right)$$

$$\stackrel{(iii)}{\leq}\sum_{i\in[n]}\left(\frac{2}{p_i}\|\mathbf{M}_i(\mathbf{z}_t-\mathbf{z}^*)\|^2+\frac{2}{p_i}\|\mathbf{M}_i(\mathbf{z}^*-\mathbf{z}_0)\|^2\right)$$

$$\leq 2(\sum_{i\in[n]}L_i)^2[\|\mathbf{z}_t-\mathbf{z}^*\|^2+\|\mathbf{z}_0-\mathbf{z}^*\|^2],$$

where we use (i) $\mathbf{M}\mathbf{z}^*=\hat{\mathbf{v}}$ by linear system condition, (ii) $\|\mathbf{z}+\mathbf{z}'\|^2\leq 2\|\mathbf{z}\|^2+2\|\mathbf{z}'\|^2$, and (iii) $\mathbb{E}\|\mathbf{z}-\mathbb{E}\mathbf{z}\|^2\leq\mathbb{E}\|\mathbf{z}\|^2$. $\square$

As a result, we can now prove the guarantee of $\texttt{AsySVRG}(\mathbf{M},\hat{\mathbf{v}},\mathbf{z}_0,\epsilon,\delta)$ as shown in Theorem 6 .

*Proof of Theorem 6.* Note we choose $\eta=\mu/(4S^2)$ and $T=8S^2/\mu^2$ as specified in Algorithm 2. Using Lemma 6 we get

$$\mathbb{E}\left\|\frac{1}{T}\sum_{t=0}^{T-1}\mathbf{z}_t-\mathbf{z}^*\right\|^2\leq\frac{2}{3}\|\mathbf{z}_0-\mathbf{z}^*\|^2.$$

For the rest of the proof, we hide constants and write $\eta=\Theta(\mu/S^2)$ and $T=\Theta(S^2/\mu^2)$ to make a constant factor progress.

Leveraging this bound we can construct the asymmetric SVRG solver by using the above variance-reduced sampling Richardson iterative process repeatedly. After $T=\Theta(S^2/\mu^2)$ iterations, we update $\mathbf{z}_0^{(q)}=\frac{1}{T}\sum_{t=0}^{T-1}\mathbf{z}_t^{(q-1)}$ and repeat the process, initalized from this average point. Consequently, after

$$Q=\log_{3/2}(\|\mathbf{z}_0-\mathbf{z}^*\|^2/\epsilon)=O(\log(\|\mathbf{z}_0-\mathbf{z}^*\|/\epsilon))$$

epochs, in expectation this algorithms returns a solution $\mathbf{z}_0^{(Q+1)}=[\sum_{t=0}^{T-1}\mathbf{z}_t^{(Q)}]/T$ satisfying

$$\mathbb{E}\|\mathbf{z}_0^{(Q+1)}-\mathbf{z}^*\|^2\leq\epsilon.$$

For runtime analysis, notice within each epoch the computational cost is one full computation of $\mathbf{Mz}$ as $\mathrm{nnz}(\mathbf{M})$ and $T$ computations of $\mathbf{M}_i\mathbf{z}$ upper bounded by $\mathcal{T}$ for each. As a result, letting $S=\sum_{i\in[n]}L_i$ the runtime in total to achieve $\|\mathbf{z}-\mathbf{z}^*\|\leq\epsilon$ is

$$O\left(\left(\mathrm{nnz}(\mathbf{M})+\mathcal{T}\cdot\frac{S^2}{\mu^2}\right)\log\left(\frac{\|\mathbf{z}_0-\mathbf{z}^*\|}{\epsilon}\right)\right).$$

Replacing now $\epsilon$ by $\epsilon\delta$ and through Markov inequality argument that

$$\mathbb{P}(\|\mathbf{z}_0^{(Q+1)}-\mathbf{z}^*\|^2\geq\epsilon)\leq\frac{\mathbb{E}[\|\mathbf{z}_0^{(Q+1)}-\mathbf{z}^*\|^2]}{\epsilon}\leq\delta,$$

we transfer this algorithm to output the desired solution in $\epsilon$-approximation with probability $1-\delta$.

□

Now we turn to discuss the acceleration runtime. Inspired by approximate proximal point [14] or Catalyst [15], and using exactly the same acceleration technique used in Palaniappan and Bach [13], when $\mathrm{nnz}(\mathbf{M})\leq\mathcal{T}(\sum_{i\in[n]}L_i)^2/\mu^2$, we can further improve this running time through the procedure as specified in Algorithm 3.

---

**Algorithm 3:** $\mathtt{AsySVRG-accelerated}(\mathbf{M},\hat{\mathbf{v}},\mathbf{z}_0,\epsilon,\delta)$

---

**Input:** $\mathbf{M}$, $\hat{\mathbf{v}}$, $\mathbf{z}_0$, $\epsilon$ desired accuracy, $\delta$ probability parameter
**Parameter:** $\tau$, $I=\tilde{O}((\tau+\mu)/\mu)$
**Output:** $\mathbf{z}^{I+1}$ as the final iterate from the outerloop
1 Initialize $\mathbf{z}^{(1)}\leftarrow\mathbf{z}_0$ **for** $i\leftarrow1$ *to* $I$ **do**
2 $\quad\left|\quad\mathbf{z}^{(i+1)}\leftarrow\right.$
$\quad\quad\mathtt{AsySVRG}(\tau\mathbf{I}+\mathbf{M},\tau\mathbf{z}^{(i)}+\hat{\mathbf{v}},\mathbf{z}^{(i)},(\frac{\mu/2}{\mu+2\tau})^2\|\mathbf{z}^{(i)}-(\tau\mathbf{I}+\mathbf{M})^{-1}(\tau\mathbf{z}^{(i)}+\hat{\mathbf{v}})\|^2,\delta/(I+1))$

---

To prove the accelerated rate, we first describe the progress per outerloop (from $z^{(i)}$ to $z^{(i+1)}$) in Algorithm 3. For the ease of analysis, we first introduce two properties of $\mathbf{M}$ as given in (8).

**Lemma 8** (Properties of $\mathbf{M}$)**.** *Let* $\mathbf{M}\in\mathbb{R}^{a\times a}$ *satisfy* $\frac{1}{2}(\mathbf{M}^\top+\mathbf{M})\succeq\mu\mathbf{I}$, *then the following properties hold true:*

1. $\|(\mathbf{I}+\mathbf{M})^{-1}\|\leq\sqrt{\frac{1}{1+2\mu}}$.

2. $\|(\tau\mathbf{M}^{-1}+\mathbf{I})^{-1}\|\leq1$, $\forall\tau\geq0$.

*Proof.* For the first property, note that

$$(\mathbf{I}+\mathbf{M})^\top(\mathbf{I}+\mathbf{M})=\mathbf{I}+(\mathbf{M}+\mathbf{M}^\top)+\mathbf{M}^\top\mathbf{M}\succeq(1+2\mu)\mathbf{I}\tag{15}$$

and therefore

$$\|(\mathbf{I}+\mathbf{M})^{-1}\|^2=\lambda_{\max}([(\mathbf{I}+\mathbf{M})^{-1}]^\top\mathbf{I}(\mathbf{I}+\mathbf{M})^{-1})\leq\frac{1}{1+2\mu}\ .$$

where we used that $\mathbf{I}\preceq(1+2\mu)^{-1}(\mathbf{I}+\mathbf{M})^\top(\mathbf{I}+\mathbf{M})$ by (15).

For the second property, when $\tau=0$ it obviously holds, so it suffices to prove for $\tau>0$. By condition we have

$$\mathbf{M}^\top\left([\mathbf{M}^{-1}]^\top+\mathbf{M}^{-1}\right)\mathbf{M}=\mathbf{M}+\mathbf{M}^\top\succeq2\mu\mathbf{I}$$

multiplying on the left by $[\mathbf{M}^{-1}]^\top$ and the right $\mathbf{M}^{-1}$ yields that

$$[\mathbf{M}^{-1}]^\top+\mathbf{M}^{-1}\succeq2\mu[\mathbf{M}^{-1}]^\top\mathbf{M}^{-1}\succeq\mathbf{0}$$

Now given $\mathbf{M}/\tau$ satisfying $\tau\mathbf{M}^{-1}+[\tau\mathbf{M}^{-1}]^\top\succeq\mathbf{0}$ and using property 1 for $\tau\mathbf{M}^{-1}$ gives the result. □

**Lemma 9** (Progress per Outerloop). *Let $\mathbf{M}$ be as in (8). Further, suppose for some $\mathbf{z}^{(0)}, \mathbf{z}^{(1)}, \hat{\mathbf{v}} \in \mathbb{R}^a$, $\tau \geq \mu$, $\epsilon \geq 0$, and $\mathbf{z}_\tau^* \stackrel{\text{def}}{=} (\tau \mathbf{I} + \mathbf{M})^{-1}(\tau \mathbf{z}^{(0)} + \hat{\mathbf{v}})$ satisfy*

$$\|\mathbf{z}^{(1)} - \mathbf{z}_\tau^*\| \leq \epsilon \|\mathbf{z}^{(0)} - \mathbf{z}_\tau^*\|.$$

*Then for $\mathbf{z}^* \stackrel{\text{def}}{=} \mathbf{M}^{-1}\hat{\mathbf{v}}$ we have*

$$\|\mathbf{z}^{(1)} - \mathbf{M}^{-1}\hat{\mathbf{v}}\| \leq \left(\frac{1}{1 + \mu/(2\tau)} + \epsilon\right)\|\mathbf{z}^{(0)} - \mathbf{M}^{-1}\hat{\mathbf{v}}\|$$

*Proof.* From the definition of $\mathbf{z}_\tau^*$ we have

$$\mathbf{z}_\tau^* - \mathbf{M}^{-1}\hat{\mathbf{v}} = (\tau\mathbf{I} + \mathbf{M})^{-1}(\tau\mathbf{z}^{(0)} + \hat{\mathbf{v}} - (\tau\mathbf{I} + \mathbf{M})\mathbf{M}^{-1}\hat{\mathbf{v}}) = \tau(\tau\mathbf{I} + \mathbf{M})^{-1}(\mathbf{z}^0 - \mathbf{M}^{-1}\hat{\mathbf{v}}) .$$

Since $\tau(\tau\mathbf{I} + \mathbf{M})^{-1} = (\mathbf{I} + \tau^{-1}\mathbf{M})$ and $\frac{1}{2}([\tau^{-1}\mathbf{M}] + [\tau^{-1}\mathbf{M}]^\top) \succeq (\mu/\tau)\mathbf{I}$ by the assumptions on $\mathbf{M}$ we have that $\|\tau(\tau\mathbf{I} + \mathbf{M})^{-1}\|_2 \leq (1 + 2\mu/\tau)^{-1/2}$ by first property in Lemma 8 and therefore

$$\|\mathbf{z}_\tau^* - \mathbf{M}^{-1}\hat{\mathbf{v}}\| \leq \sqrt{\frac{1}{1 + 2\mu/\tau}}\|\mathbf{z}^{(0)} - \mathbf{M}^{-1}\hat{\mathbf{v}}\| \leq \frac{1}{1 + \mu/(2\tau)}\|\mathbf{z}^{(0)} - \mathbf{M}^{-1}\hat{\mathbf{v}}\| , \qquad (16)$$

where the last inequality follows from the fact $\sqrt{\frac{1}{1+2\mu/\tau}} \leq \frac{1}{1+\mu/(2\tau)}$ for all $\mu \leq \tau$. Further we have that

$$\mathbf{z}^{(0)} - \mathbf{z}_\tau^* = (\tau\mathbf{I} + \mathbf{M})^{-1}((\tau\mathbf{I} + \mathbf{M})\mathbf{z}^0 - \tau\mathbf{z}^0 - \mathbf{M}\mathbf{M}^{-1}\hat{\mathbf{v}}) = (\tau\mathbf{M}^{-1} + \mathbf{I})^{-1}(\mathbf{z}^{(0)} - \mathbf{M}^{-1}\hat{\mathbf{v}}) .$$

by the second property of Lemma 8 we have that $\|(\tau\mathbf{M}^{-1} + \mathbf{I})^{-1}\|_2 \leq 1$ and

$$\|\mathbf{z}^{(0)} - (\tau\mathbf{I} + \mathbf{M})^{-1}(\tau\mathbf{z}^0 + \hat{\mathbf{v}})\| \leq \|\mathbf{z}^{(0)} - \mathbf{M}^{-1}\hat{\mathbf{v}}\| . \qquad (17)$$

As $\|z^{(1)} - \mathbf{M}^{-1}\hat{v}\| \leq \|\mathbf{z}^{(1)} - \mathbf{z}_\tau^*\| + \|\mathbf{z}_\tau^* + \mathbf{M}^{-1}\hat{\mathbf{v}}\|$ the result follows by (16) and (17). $\square$

*Proof of Theorem 7.* The acceleration runtime can be achieved through a standard outer acceleration procedure:

Denote the whole optimizer $\mathbf{z}^*$ satisfying $\mathbf{M}\mathbf{z}^* = \hat{\mathbf{v}}$, $S = \sum_{i \in [n]} L_i$ as usual. When $\mathrm{nnz}(\mathbf{M}) \leq \mathcal{T} \cdot S^2/\mu^2$, we choose $\tau = S\sqrt{\mathcal{T}/\mathrm{nnz}(\mathbf{M})} \geq \mu$.

Using a similar derivation as in Theorem 6 we know after $T = \tilde{O}(\cdot(\mathrm{nnz}(\mathbf{M}) + \mathcal{T}(S + \tau)^2/(\mu + \tau)^2))$, we have $\mathbf{z}^{(i+1)}$ satisfying with probability at least $1 - \delta/(I+1)$

$$\|\mathbf{z}^{(i+1)} - (\tau\mathbf{I} + \mathbf{M})^{-1}(\tau\mathbf{z}^{(i)} + \hat{\mathbf{v}})\|^2 \leq \left(\frac{\mu/2}{\mu + 2\tau}\right)^2 \|\mathbf{z}^{(i)} - (\tau\mathbf{I} + \mathbf{M})^{-1}(\tau\mathbf{z}^{(i)} + \hat{\mathbf{v}})\|^2.$$

Starting the induction from $i = 0$ and using Lemma 9 (since we have $\tau \geq \mu$) recursively, it implies with probability at least $1 - (i+1)/(I+1) \cdot \delta$,

$$\|\mathbf{z}^{(i+1)} - \mathbf{z}^*\| \leq \left(\frac{1}{1 + \mu/2\tau} + \frac{\mu/2}{\mu + 2\tau}\right)\|\mathbf{z}^{(i)} - \mathbf{z}^*\|,$$

$$\|\mathbf{z}^{(i+1)} - \mathbf{z}^*\| \leq \left(\frac{1}{1 + \mu/2\tau} + \frac{\mu/2}{\mu + 2\tau}\right)^{i+1}\|\mathbf{z}^{(0)} - \mathbf{z}^*\|.$$

Note as we choose $I = \tilde{O}((\tau + \mu)/\mu)$ we have with probability $1 - \delta$ after $I$ outerloops,

$$\|\mathbf{z}^{(I+1)} - \mathbf{z}^*\| \leq \epsilon,$$

which takes a total runtime of

$$\tilde{O}\left(\left(\frac{\mu + \tau}{\mu}\right)\left(\mathrm{nnz}(\mathbf{M}) + \mathcal{T}\frac{(S + \tau)^2}{(\mu + \tau)^2}\right)\right) = \tilde{O}\left(\sqrt{\mathrm{nnz}(\mathbf{M})\mathcal{T}}\frac{\sum_{i \in [n]} L_i}{\mu}\right).$$

$\square$

## C.2 Proofs of Results in Section 3.2

To solve the particular asymmetric system we consider the step

$$\mathbf{z}_{t+1} = \mathbf{z}_t - \frac{\eta}{p_i}\big(\mathbf{M}_i\mathbf{z}_t - \mathbf{M}_i\mathbf{z}_0 + p_i(\mathbf{M}\mathbf{z}_0 - \hat{\mathbf{v}})\big)$$

$$\text{where } \mathbf{M}_i \overset{\text{def}}{=} \begin{pmatrix} \frac{\|\mathbf{a}_i\|^2}{\|\mathbf{A}\|_F^2}\mathbf{I} & -\frac{1}{\mu}(\mathbf{a}_i\mathbf{a}_i^\top - c\frac{\|\mathbf{a}_i\|^2}{\|\mathbf{A}\|_F^2}\mathbf{I}) \\ \frac{1}{\mu}(\mathbf{a}_i\mathbf{a}_i^\top - c\frac{\|\mathbf{a}_i\|^2}{\|\mathbf{A}\|_F^2})\mathbf{I} & \frac{\|\mathbf{a}_i\|^2}{\|\mathbf{A}\|_F^2}\mathbf{I} \end{pmatrix}, \; p_i \propto \|\mathbf{a}_i\|^2, \; \forall i \in [n]. \tag{18}$$

Through a more fine-grained analysis, $\texttt{AsySVRG}(\mathbf{M},\hat{\mathbf{v}},\mathbf{z}_0,\epsilon,\delta)$ with particular choices of $\eta$, $T$, $\mathbf{M}_i$, and $\{p_i\}_{i\in[n]}$, can have a better runtime guarantee and be accelerated using a similar idea as in the general case discussed in previous subsection. This is stated formally using the following variance bound in Theorem 8.

**Lemma 10** (Variance bound for specific form). *For problem* (10), *the variance incurred in* (18) *is bounded by* $S = O(\|\mathbf{A}\|_F\sqrt{\lambda_1}/\mu)$, *i.e. there exists constant* $0 < C < \infty$ *that*

$$\sum_{i\in[n]}\frac{1}{p_i}\|\mathbf{M}_i\mathbf{z}_t - \mathbf{M}_i\mathbf{z}_0 + p_i\cdot(\mathbf{M}\mathbf{z}_0 - \hat{\mathbf{v}})\|^2 \le \frac{C\|\mathbf{A}\|_F^2\lambda_1}{\mu^2}\big[\|\mathbf{z}_t - \mathbf{z}^*\|^2 + \|\mathbf{z}_t - \mathbf{z}^*\|^2\big].$$

*Proof.* For arbitrary $\Delta \in \mathbb{R}^{2d}$, set $p_i = \|\mathbf{a}_i\|^2/\|\mathbf{A}\|_F^2$, we have

$$\sum_{i\in[n]}\frac{1}{p_i}\|\mathbf{M}_i\Delta\|^2$$

$$= \Delta^\top\Big(\sum_{i\in[n]}\frac{1}{p_i}\mathbf{M}_i^\top\mathbf{M}_i\Big)\Delta$$

$$\overset{(i)}{=} \Delta^\top\left(\sum_{i\in[n]}\frac{1}{p_i}\begin{pmatrix} p_i^2\mathbf{I} + \frac{1}{\mu^2}(\mathbf{a}_i\mathbf{a}_i^\top - p_icI)^2 & \mathbf{0} \\ \mathbf{0} & \frac{1}{\mu^2}(\mathbf{a}_i\mathbf{a}_i^\top - p_icI)^2 + p_i^2\mathbf{I} \end{pmatrix}\right)\Delta$$

$$\overset{(ii)}{=} \Delta^\top\begin{pmatrix} \mathbf{I} + \frac{1}{\mu^2}(\|\mathbf{A}\|_F^2\mathbf{A}^\top\mathbf{A} - 2c\mathbf{A}^\top\mathbf{A} + c^2\mathbf{I}) & \mathbf{0} \\ \mathbf{0} & \mathbf{I} + \frac{1}{\mu^2}(\|\mathbf{A}\|_F^2\mathbf{A}^\top\mathbf{A} - 2c\mathbf{A}^\top\mathbf{A} + c^2\mathbf{I}) \end{pmatrix}\Delta$$

$$\overset{(iii)}{\le} \Big(1 + \frac{\|\mathbf{A}\|_F^2\lambda_1}{\mu^2}\Big)\|\Delta\|^2 \le \frac{C\|\mathbf{A}\|_F^2\lambda_1}{\mu^2}\|\Delta\|^2,$$

where we use (i) the specific form of $\mathbf{M}_i$ as in (18), (ii) specific choice of $p_i = \|\mathbf{a}_i\|^2/\|\mathbf{A}\|_F^2$, (iii) the assumption that $\mathbf{A}^\top\mathbf{A} \preceq \lambda_1\mathbf{I}$ and $c \in [0, \lambda_1]$.

Note then similar to proof of Lemma 7, we have

$$\sum_{i\in[n]}\frac{1}{p_i}\|\mathbf{M}_i(\mathbf{z}_t - \mathbf{z}_0) + p_i(\mathbf{M}\mathbf{z}_0 - \mathbf{v})\|^2 \le \sum_{i\in[n]}\Big(\frac{2}{p_i}\|\mathbf{M}_i(\mathbf{z}_t - \mathbf{z}^*)\|^2 + \frac{2}{p_i}\|\mathbf{M}_i(\mathbf{z}^* - \mathbf{z}_0)\|^2\Big)$$

$$\le \frac{2C\|\mathbf{A}\|_F^2\lambda_1}{\mu^2}(\|\mathbf{z}_t - \mathbf{z}^*\|^2 + \|\mathbf{z}_0 - \mathbf{z}^*\|^2).$$

$\square$

*Proof of Theorem 8.* Using Lemma 10, the conditions of Lemma 6 are satisfied with $S = \sqrt{C}\|\mathbf{A}\|_F\sqrt{\lambda_1}/\mu$. Consequently we have

$$\mathbb{E}\Big\|\frac{1}{T}\sum_{t=0}^{T-1}\mathbf{z}_t - \mathbf{z}^*\Big\|^2 \le \mathbb{E}\Big[\frac{1}{T}\sum_{t=0}^{T-1}\|\mathbf{z}_t - \mathbf{z}^*\|^2\Big] \le \frac{\frac{1}{2T} + \eta^2C\|\mathbf{A}\|_F^2\lambda_1/\mu^2}{\eta - \eta^2C\|\mathbf{A}\|_F^2\lambda_1/\mu^2}\|\mathbf{z}_0 - \mathbf{z}^*\|^2. \tag{19}$$

**For the non-acceleration case:** Note we choose $\eta = \mu^2/(C\|\mathbf{A}\|_F^2\lambda_1)$, and $T = C\|\mathbf{A}\|_F^2\lambda_1/\mu^2$ as in Eq. (19) yields

$$\mathbb{E}\left\|\frac{1}{T}\sum_{t=0}^{T-1}\mathbf{z}_t - \mathbf{z}^*\right\|^2 \leq \frac{2}{3}\|\mathbf{z}_0 - \mathbf{z}^*\|^2.$$

Hereinafter, we hide constants and write $\eta = \Theta(\mu^2/(\|\mathbf{A}\|_F^2\lambda_1))$ and $T = O(\|A\|_F^2\lambda_1/\mu^2)$ to make a constant factor progress.

Then similarly as in Algorithm 2 and Theorem 6, we argue after $Q = O(\log(\|\mathbf{z}_0 - \mathbf{z}^*\|/\epsilon))$ batches, in expectation we'll return a solution $\mathbf{z}_0^{(Q+1)} = [\sum_{t=0}^{T-1}\mathbf{z}_t^{(Q)}]/T$

$$\mathbb{E}\|\mathbf{z}_0^{(Q+1)} - \mathbf{z}^*\|^2 \leq \epsilon.$$

For runtime analysis, notice within each batch the computational cost is one full computation of $\mathbf{Mz}$ and $O(T)$ computations of $\mathbf{M}_i\mathbf{z}$, which together is $O(\mathrm{nnz}(\mathbf{A}) + d\cdot\|\mathbf{A}\|_F^2\lambda_1/\mu^2)$. So the total runtime to achieve $\|\mathbf{z}^{(Q+1)} - \mathbf{z}^*\| \leq \epsilon$ with probability $1 - \delta$ is

$$O\big((\mathrm{nnz}(\mathbf{A}) + d\cdot\|\mathbf{A}\|_F^2\lambda_1/\mu^2)\log(\|\mathbf{z}_0 - \mathbf{z}^*\|/\epsilon\delta)\big) = \tilde{O}(\mathrm{nnz}(\mathbf{A}) + d\cdot\mathrm{sr}(\mathbf{A})\lambda_1^2/\mu^2).$$

**For the acceleration case:** The standard technique of outer acceleration used in Theorem 7 is applied to get a better runtime under this case, and is used to prove Theorem 3.

$\square$

### C.3 Squared System Solver Using SVRG

For the squared ridge regression solver, we first give its simple pseudocode using Lemma 3 and `AsySVRG` for completeness.

---

**Algorithm 4:** `RidgeSquare(`$\mathbf{A}, c, \mu^2, \mathbf{v}, \epsilon, \delta$`)`

**Input:** $\mathbf{A}$ data matrix, $c \in [0, \lambda_1]$, $\mu > 0$, $\mathbf{v}$, $\mathbf{x}_0$ initial, $\epsilon$ accuracy, $\delta$ probability
**Output:** $\widetilde{\mathbf{x}}$ $\epsilon$-approximate solution as in Definition 3.
1  Initialize $\mathbf{z}^0$.
2  Set $\mathbf{M} = \begin{pmatrix} \mathbf{I} & -\frac{1}{\mu}(\mathbf{A}^\top\mathbf{A} - c\mathbf{I}) \\ \frac{1}{\mu}(\mathbf{A}^\top\mathbf{A} - c\mathbf{I}) & \mathbf{I} \end{pmatrix}$ and $\hat{\mathbf{v}} = (\mathbf{0}, \mathbf{v}/\mu^2)^\top$.
3  Call $[\mathbf{x}, \mathbf{y}] \leftarrow$ `AsySVRG(`$\mathbf{M}, \hat{\mathbf{v}}, \mathbf{z}_0, \epsilon\|\mathbf{v}\|, \delta$`)`.
4  Return $\widetilde{\mathbf{x}} \leftarrow \mathbf{y}$

---

This is essentially a corollary of Theorem 8.

*Proof of Theorem 3.* Set

$$\mathbf{M} = \begin{pmatrix} \mathbf{I} & -\frac{1}{\mu}(\mathbf{A}^\top\mathbf{A} - c\mathbf{I}) \\ \frac{1}{\mu}(\mathbf{A}^\top\mathbf{A} - c\mathbf{I}) & \mathbf{I} \end{pmatrix} \text{ and } \hat{\mathbf{v}} = (\mathbf{0}, \mathbf{v}/\mu^2)^\top, c \in [0, \lambda_1].$$

We know an $\epsilon$-approximate squared ridge regression solver would suffice to call

$$[\mathbf{x}, \mathbf{y}] \leftarrow \texttt{AsySVRG}(\mathbf{M}, \hat{\mathbf{v}}, \mathbf{z}_0, \epsilon\|\mathbf{v}\|, \delta).$$

once and set its output as $\mathbf{y}$ through Lemma 3. This together with Theorem 8 gives us the total runtime of $\tilde{O}\big(\mathrm{nnz}(\mathbf{A}) + d\cdot\mathrm{sr}(\mathbf{A})\lambda_1^2/\mu^2\big)$ unaccelerated and $\tilde{O}\big(\sqrt{\mathrm{nnz}(\mathbf{A})d\cdot\mathrm{sr}(\mathbf{A})}\lambda_1/\mu\big)$ accelerated (when $\mathrm{nnz}(\mathbf{A}) \leq d\cdot\mathrm{sr}(\mathbf{A})\lambda_1^2/\mu^2$). In short, we have the guaranteed running time within

$$\tilde{O}\big(\mathrm{nnz}(\mathbf{A}) + \sqrt{\mathrm{nnz}(\mathbf{A})d\cdot\mathrm{sr}(\mathbf{A})}\lambda_1/\mu\big).$$

$\square$

Theorem 4 also implies immediately a solver for non-PSD system in form: $(\mathbf{A}^\top\mathbf{A} - c\mathbf{I})\mathbf{x} = \mathbf{v}$, $c \in [0, \lambda_1]$ with same runtime guarantee whenever all eigenvalues $\lambda_i - c$ of $(\mathbf{A}^\top\mathbf{A} - c\mathbf{I})$ satisfy $|\lambda_i - c| \geq \mu > 0, \forall i$. This is done by considering solving $(\mathbf{A}^\top\mathbf{A} - c\mathbf{I})^2\mathbf{x} = (\mathbf{A}^\top\mathbf{A} - c\mathbf{I})\mathbf{v}$. We state this formally in the following corollary for completeness, which is equivalent as showing Corollary 1.

**Corollary 2.** *Given $c \in [0, \lambda_1]$, and a non-PSD system $(\mathbf{A}^\top \mathbf{A} - c\mathbf{I})\mathbf{x} = \mathbf{v}$ and an initial point $\mathbf{x}_0$, for arbitrary $c \in \mathbb{R}$ satisfying $(\mathbf{A}^\top \mathbf{A} - c\mathbf{I})^2 \succeq \mu^2 \mathbf{I}, \mu > 0$, there is an algorithm that uses $\mathtt{AsySVRG}(\mathbf{M}, \hat{\mathbf{v}}, \mathbf{z}_0, \Theta(\epsilon\|\mathbf{v}\|), \delta)$ to return with probablity $1 - \delta$ a solution $\widetilde{\mathbf{x}}$ such that $\|\widetilde{\mathbf{x}} - (\mathbf{A}^\top \mathbf{A} - c\mathbf{I})^{-1}\mathbf{v}\| \leq \epsilon\|\mathbf{v}\|$, within runtime $\tilde{O}\big(\mathrm{nnz}(\mathbf{A}) + d \cdot \mathrm{sr}(\mathbf{A})\lambda_1^2/\mu\big)$. The runtime can be accelerated to $\tilde{O}\big(\lambda_1 \sqrt{\mathrm{nnz}(\mathbf{A})d \cdot \mathrm{sr}(\mathbf{A})}/\mu\big)$ when $\mathrm{nnz}(\mathbf{A}) \leq d \cdot \mathrm{sr}(\mathbf{A})\lambda_1^2/\mu^2$.*

*Proof.* It suffices to show that we can solve $(\mathbf{A}^\top \mathbf{A} - c\mathbf{I})^2 \mathbf{x} = (\mathbf{A}^\top \mathbf{A} - c\mathbf{I})\mathbf{v}$ to high accuracy within the desired runtime for any given $\mathbf{v}$.

By Theorem 3, whenever $(\mathbf{A}^\top \mathbf{A} - c\mathbf{I})^2 \succeq \mu^2 \mathbf{I}$, we can solve $\big((\mathbf{A}^\top \mathbf{A} - c\mathbf{I})^2 + \mu^2 \mathbf{I}\big)\mathbf{x} = (\mathbf{A}^\top \mathbf{A} - c\mathbf{I})\mathbf{v}$ to $\epsilon$-approximate accuracy within runtime $\tilde{O}(\mathrm{nnz}(\mathbf{A}) + \sqrt{\mathrm{nnz}(\mathbf{A})d \cdot \mathrm{sr}(\mathbf{A})}\lambda_1/\mu)$.

Now we can consider preconditioning $(\mathbf{A}^\top \mathbf{A} - c\mathbf{I})^2$ using $(\mathbf{A}^\top \mathbf{A} - c\mathbf{I})^2 + \mu^2 \mathbf{I}$ by noticing that $(\mathbf{A}^\top \mathbf{A} - c\mathbf{I})^2 + \mu^2 \mathbf{I} \approx_{1/2} (\mathbf{A}^\top \mathbf{A} - c\mathbf{I})^2$ under the condition.

As a result, it suffices to apply Richardson update

$$x^{(t+1)} \leftarrow x^{(t)} - \eta \big[(\mathbf{A}^\top \mathbf{A} - c\mathbf{I})^2 + \mu^2 \mathbf{I}\big]^{-1}\Big((\mathbf{A}^\top \mathbf{A} - c\mathbf{I})^2 x^{(t)} - (\mathbf{A}^\top \mathbf{A} - c\mathbf{I})\mathbf{v}\Big),$$

with $\eta = 1/2$. Since it satisfies $\frac{1}{2}\mathbf{I} \preceq \big[(\mathbf{A}^\top \mathbf{A} - c\mathbf{I})^2 + \mu^2 \mathbf{I}\big]^{-1}(\mathbf{A}^\top \mathbf{A} - c\mathbf{I})^2 \preceq \mathbf{I}$, it achieves $\epsilon$ accuracy $\|x^{(T)} - (\mathbf{A}^\top \mathbf{A} - c\mathbf{I})^{-2}\mathbf{v}\| \leq \epsilon\|\mathbf{v}\|$ in

$$O\left(\log\left(\frac{\|\mathbf{x}_0 - (\mathbf{A}^\top \mathbf{A} - c\mathbf{I})^{-1}\mathbf{v}\|}{\epsilon\|\mathbf{v}\|}\right)\right) = \tilde{O}(1)$$

iterations, with each iteration cost $\tilde{O}\big(\mathrm{nnz}(\mathbf{A}) + d \cdot \mathrm{sr}(\mathbf{A})\lambda_1^2/\mu^2\big)$ using the unaccelerated subroutine, and $\tilde{O}\big(\sqrt{d \cdot \mathrm{sr}(\mathbf{A})}\lambda_1/\mu\big)$ using the accelerated subroutine when $\mathrm{nnz}(\mathbf{A}) \leq d \cdot \mathrm{sr}(\mathbf{A})\lambda_1^2/\mu^2$. This leads to a total unaccelerated runtime of $\tilde{O}\big(\mathrm{nnz}(\mathbf{A}) + d \cdot \mathrm{sr}(\mathbf{A})\lambda_1^2/\mu\big)$, and accelerated runtime $\tilde{O}\big(\lambda_1 \sqrt{\mathrm{nnz}(\mathbf{A})d \cdot \mathrm{sr}(\mathbf{A})}/\mu\big)$ when $\mathrm{nnz}(\mathbf{A}) \leq d \cdot \mathrm{sr}(\mathbf{A})\lambda_1^2/\mu^2$. $\qquad\square$

In the end of this section, we remark that all proofs and results are stated without the condition $\lambda_1 \leq 1$ to give a clearer sense of the runtime dependence in general setting. When applied to our specific case of solving squared systems, due to renormalization we have $\lambda_1 \in [1/2, 1]$ and $\lambda \leftarrow \lambda/\lambda_1 = 1/\kappa$ which lead to running times formally stated in Theorems 1 to 3 and proved in Appendix D.

### C.4 Direct Methods Runtime

We first state the results and corresponding proofs of using direct methods stated in the beginning of Section 3. Consider the squared system $\big((\mathbf{A}^\top \mathbf{A} - c\mathbf{I})^2 + \mu^2 \mathbf{I}\big)\mathbf{x} = \mathbf{v}$ for given $\mathbf{A} \in \mathbb{R}^{n \times d}, \mathbf{v} \in \mathbb{R}^d$, $\mu > 0$ and $c \in [0, \lambda_1]$.

The following theorem gives a bound of the running time of solving this system using accelerated gradient descent [29, 30].

**Theorem 10** (Direct AGD Runtime). *Consider iteration*

$$\mathbf{x}_{t+1} = \mathbf{x}_t - \gamma\big[\big((\mathbf{A}^\top \mathbf{A} - c\mathbf{I})^2 + \mu^2 \mathbf{I}\big)\mathbf{x}_t - \mathbf{v}\big] + \beta(\mathbf{x}_t - \mathbf{x}_{t-1}).$$

*Choosing $\gamma = \frac{4}{\lambda_1 + 2\mu}, \beta = \frac{\lambda_1}{\lambda_1 + 2\mu}$ under above condition, the total running time to get an $\epsilon$-approximate solution is $\tilde{O}(\mathrm{nnz}(\mathbf{A})\lambda_1/\mu)$.*

Now we turn to analyzing SVRG directly applied to the squared system as follows

$$\big((\mathbf{A}^\top \mathbf{A} - c\mathbf{I})^2 + \mu^2 \mathbf{I}\big)\mathbf{x} = \mathbf{v}. \tag{20}$$

We take the step

$$\mathbf{x}_{t+1} = \mathbf{x}_t - \frac{\eta}{p_{ij}}\big(\mathbf{M}_{ij}\mathbf{x}_t - \mathbf{M}_{ij}\mathbf{x}_0 + p_{ij}(\mathbf{M}\mathbf{x}_0 - \mathbf{v})\big)$$

$$\text{where } \mathbf{M}_{ij} \stackrel{\mathrm{def}}{=} \mathbf{a}_i \mathbf{a}_i^\top \mathbf{a}_j \mathbf{a}_j^\top - 2c\frac{\|\mathbf{a}_j\|^2}{\|\mathbf{A}\|_{\mathrm{F}}^2}\mathbf{a}_i \mathbf{a}_i^\top + (c^2 + \mu^2)\frac{\|\mathbf{a}_j\|^2\|\mathbf{a}_i\|^2}{\|\mathbf{A}\|_{\mathrm{F}}^4}\mathbf{I} \tag{21}$$

$$\text{and } p_{ij} \propto \|\mathbf{a}_i\|^2 \|\mathbf{a}_j\|^2, \forall i, j \in [n].$$

Such update gives the following variance bound.

**Lemma 11** (Variance bound for solving squared system directly). *For problem* (20), *the variance incurred in* (21) *is bounded by* $S = O((\lambda_1^2\|\mathbf{A}\|_F^4 + (\lambda_1^2 + \mu^2)^2)/\mu^2)$, *i.e. there exists* $0 < C < \infty$ *that*

$$\sum_{i,j\in[n]} \frac{1}{p_{ij}} \|\mathbf{M}_{ij}\mathbf{x}_t - \mathbf{M}_{ij}\mathbf{x}_0 + p_{ij}(\mathbf{M}\mathbf{x}_0 - \mathbf{v})\|^2 \leq \frac{C(\lambda_1^2\|\mathbf{A}\|_F^4 + (\lambda_1^2 + \mu^2)^2)}{\mu^2}[\|\mathbf{x}_t - \mathbf{x}^*\|^2 + \|\mathbf{x}_0 - \mathbf{x}^*\|^2],$$

*where* $C < \infty$ *is a numerical constant and* $\|A\|_F^2 \leq d^2$.

*Proof.* Notice that $\nabla\psi_{ij} = [\mathbf{a}_i\mathbf{a}_i^\top\mathbf{a}_j\mathbf{a}_j^\top - 2c\frac{\|\mathbf{a}_j\|^2}{\|\mathbf{A}\|_F^2}\mathbf{a}_i\mathbf{a}_i^\top + (c^2 + \mu^2)\frac{\|\mathbf{a}_j\|^2\|\mathbf{a}_i\|^2}{\|\mathbf{A}\|_F^4}\mathbf{I}]\mathbf{x} - \frac{\|\mathbf{a}_j\|^2\|\mathbf{a}_i\|^2}{\|\mathbf{A}\|_F^4}\mathbf{v}$, by bounding directly and summing up all terms $i,j \in [n]$, we get

$$\sum_{i,j\in[n]} \frac{1}{p_{ij}} \|\mathbf{M}_{ij}\mathbf{x} - \mathbf{M}_{ij}\mathbf{x}^*\|^2$$

$$= (\mathbf{x} - \mathbf{x}^*)^\top \left(\sum_{i,j\in[n]} \frac{1}{p_{ij}}\mathbf{M}_{ij}^\top\mathbf{M}_{ij}\right)(\mathbf{x} - \mathbf{x}^*)$$

$$= \sum_{i,j\in[n]} \frac{\|\mathbf{A}\|_F^4}{\|\mathbf{a}_i\|^2\|\mathbf{a}_j\|^2}\|\left(\mathbf{a}_i\mathbf{a}_i^\top\mathbf{a}_j\mathbf{a}_j^\top - 2c\frac{\|\mathbf{a}_j\|^2}{\|\mathbf{A}\|_F^2}\mathbf{a}_i\mathbf{a}_i^\top + \frac{\|\mathbf{a}_i\|^2\|\mathbf{a}_j\|^2}{\|\mathbf{A}\|_F^4}(c^2 + \mu^2)I\right)(\mathbf{x} - \mathbf{x}^*)\|^2$$

$$= (\mathbf{x} - \mathbf{x}^*)^\top \left(\sum_{i,j\in[n]} \left(\frac{\|\mathbf{A}\|_F^4}{\|\mathbf{a}_j\|^2}\mathbf{a}_j\mathbf{a}_j^\top\mathbf{a}_i\mathbf{a}_i^\top\mathbf{a}_j\mathbf{a}_j^\top + 4c^2\|\mathbf{a}_j\|^2\mathbf{a}_i\mathbf{a}_i^\top + (c^2 + \mu^2)^2\frac{\|\mathbf{a}_i\|^2\|\mathbf{a}_j\|^2}{\|\mathbf{A}\|_F^4}\mathbf{I}\right.\right.$$

$$- 2c\|\mathbf{A}\|_F^2(\mathbf{a}_j\mathbf{a}_j^\top\mathbf{a}_i\mathbf{a}_i^\top + \mathbf{a}_i\mathbf{a}_i^\top\mathbf{a}_j\mathbf{a}_j^\top) + (c^2 + \mu^2)(\mathbf{a}_j\mathbf{a}_j^\top\mathbf{a}_i\mathbf{a}_i^\top + \mathbf{a}_i\mathbf{a}_i^\top\mathbf{a}_j\mathbf{a}_j^\top)$$

$$\left.\left. - 4c(c^2 + \mu^2)\frac{\|\mathbf{a}_j\|^2}{\|\mathbf{A}\|_F^2}\mathbf{a}_i\mathbf{a}_i^\top\right)\right)(\mathbf{x} - \mathbf{x}^*)$$

$$= (\mathbf{x} - \mathbf{x}^*)^\top \left(\|\mathbf{A}\|_F^4(\mathbf{A}^\top\mathbf{A})(\mathbf{A}^\top\mathbf{A}) + 4c^2\|\mathbf{A}\|_F^2(\mathbf{A}^\top\mathbf{A}) + (c^2 + \mu^2)^2\mathbf{I}\right.$$

$$\left. - 4c^2(\mathbf{A}^\top\mathbf{A})(\mathbf{A}^\top\mathbf{A}) + 2(c^2 + \mu^2)(\mathbf{A}^\top\mathbf{A})(\mathbf{A}^\top\mathbf{A}) - 4c(c^2 + \mu^2)(\mathbf{A}^\top\mathbf{A})\right)(\mathbf{x} - \mathbf{x}^*)$$

$$\overset{(i)}{\leq} (\mathbf{x} - \mathbf{x}^*)^\top \left(\|\mathbf{A}\|_F^4\lambda_1^2\mathbf{I} + 4c^2\lambda_1\|\mathbf{A}\|_F^2\mathbf{I} + (c^2 + \mu^2)^2\mathbf{I} + 2(c^2 + \mu^2)\lambda_1^2\right)(\mathbf{x} - \mathbf{x}^*)$$

$$\leq C\left(\lambda_1^2\|\mathbf{A}\|_F^4 + 4c^2\lambda_1\|\mathbf{A}\|_F^2 + (c^2 + \mu^2)(c^2 + \mu^2 + 2\lambda_1^2)\right)\|\mathbf{x} - \mathbf{x}^*\|^2$$

$$\overset{(ii)}{\leq} C\left(\lambda_1^2\|\mathbf{A}\|_F^4 + (\lambda_1^2 + \mu^2)^2\right)\|\mathbf{x} - \mathbf{x}^*\|^2,$$

where we use $\mathbf{A}^\top\mathbf{A} \preceq \lambda_1\mathbf{I}$ and $c \in [0, \lambda_1]$ in (i) by condition, and (ii) holds for some constant $0 < C < \infty$. Then we conclude from the fact that $\|\mathbf{x} + \mathbf{x}'\|^2 \leq 2\|\mathbf{x}\|^2 + 2\|\mathbf{x}'\|^2$ and $\mathbb{E}\|\mathbf{x} - \mathbb{E}\mathbf{x}\|^2 \leq \mathbb{E}\|\mathbf{x}\|^2$ that

$$\sum_{i,j\in[n]} \frac{1}{p_{ij}} \|\mathbf{M}_{ij}\mathbf{x}_t - \mathbf{M}_{ij}\mathbf{x}_0 + p_{ij}(\mathbf{M}\mathbf{x}_0 - \mathbf{v})\|^2 \leq \frac{C(\lambda_1^2\|\mathbf{A}\|_F^4 + (\lambda_1^2 + \mu^2)^2)}{\mu^2}[\|\mathbf{x}_t - \mathbf{x}^*\|^2 + \|\mathbf{x}_0 - \mathbf{x}^*\|^2].$$

$\square$

**Theorem 11** (Direct SVRG Runtime). *For problem* (20), *the SVRG algorithm applied with* (21) *returns with probability* $\geq 1 - \delta$ *an* $\epsilon$-*approximate solution in* $\tilde{O}(\mathrm{nnz}(\mathbf{A}) + d \cdot \mathrm{sr}(\mathbf{A})^2\lambda_1^4/\mu^4)$ *time. An accelerated variant of it improves the runtime to* $\tilde{O}(\mathrm{nnz}(\mathbf{A}) + \mathrm{nnz}(\mathbf{A})^{3/4}d^{1/4}\mathrm{sr}(\mathbf{A})^{1/2}\lambda_1/\mu)$.

The proof is a direct combination of Lemma 6 with the proving technique for Theorem 8. We omit it here as the procedure and argument are basically the same.

From that we can get a direct SVRG solver of squared system $((\mathbf{A}^\top\mathbf{A} - c\mathbf{I})^2 + \mu^2\mathbf{I})\mathbf{x} = \mathbf{v}$ that outputs $\epsilon$-approximate solution with high probability with running time $\tilde{O}(\mathrm{nnz}(\mathbf{A}) + d \cdot \mathrm{sr}(\mathbf{A})^2\lambda_1^4/\mu^4)$.

Because of the high complexity of above methods (either not nearly-linear in AGD or squaring problem dimension, i.e. having $\mathrm{sr}(\mathbf{A})^2\lambda_1^4/\mu^4$ as condition number in SVRG), a new insight is required to better solve such systems. The technique we develop for this purpose is to 'decouple' the squared matrix at the cost of asymmetry, formally introduced by the following reduction.

## D   Proofs for Main Results

In this section, we prove our main theorems for new algorithms on PCP and PCR problems stated in Section 1.3. As a byproduct of the results, we can also use some variant of the Zolotarev rational to approximate the square-root function, and thus build efficient square-root-matrix-and-vector solver (see Theorem 4 and Algorithm 7).

To begin with, we introduce a helper lemma useful for analyzing the approximation property of the theorem.

**Lemma 12** (Accumulative Error from Products). *If there are procedures $\mathcal{C}_i(\mathbf{v}), i \in [k]$ that carries out a product computation $\mathbf{C}_i \cdot \mathbf{v}$ in $\epsilon$ accuracy, i.e. $\forall i \in [k], \|\mathcal{C}_i(\mathbf{v}) - \mathbf{C}_i\mathbf{v}\| \leq \epsilon\|\mathbf{v}\|$, and that $\|\mathbf{C}_i\| \leq M, \forall i \in [k]$ for some $M \geq 1$. When $\epsilon \leq M/2k$, we have*

$$\|\mathcal{C}_k(\mathcal{C}_{k-1}(\cdots\mathcal{C}_1(\mathbf{v}))) - \prod_{i=1}^{k}\mathbf{C}_i\mathbf{v}\| \leq 2\epsilon k M^{k-1}\|\mathbf{v}\|.$$

*Proof.* By induction, for $k=1$, this is true since $\|\mathcal{C}_1(\mathbf{v}) - \mathbf{C}_1\mathbf{v}\| \leq \epsilon\|\mathbf{v}\|$. Suppose this is true for $i$, i.e. $\|\mathcal{C}_i(\mathcal{C}_{i-1}(\cdots\mathcal{C}_1(\mathbf{v}))) - \prod_{j=1}^{i}\mathbf{C}_j\mathbf{v}\| \leq 2\epsilon i M^{i-1}\|\mathbf{v}\|$, then for $i+1$, we have

$$\|\mathcal{C}_{i+1}(\mathcal{C}_i(\cdots\mathcal{C}_1(\mathbf{v}))) - \prod_{j=1}^{i+1}\mathbf{C}_j\mathbf{v}\| \leq \|\mathcal{C}_{i+1}(\mathcal{C}_i(\cdots\mathcal{C}_1(\mathbf{v}))) - \mathbf{C}_{i+1}\mathcal{C}_i(\cdots\mathcal{C}_1(\mathbf{v}))\|$$

$$+ \|\mathbf{C}_{i+1}(\mathcal{C}_i(\cdots\mathcal{C}_1(\mathbf{v})) - \prod_{j=1}^{i}\mathbf{C}_j\mathbf{v})\|$$

$$\leq \epsilon\|\mathcal{C}_i(\cdots\mathcal{C}_1(\mathbf{v}))\| + \|\mathbf{C}_{i+1}\|\|\mathcal{C}_i(\cdots\mathcal{C}_1(\mathbf{v})) - \prod_{j=1}^{i}\mathbf{C}_j\mathbf{v}\|$$

$$\leq \epsilon(M^i\|\mathbf{v}\| + 2\epsilon i M^{i-1}\|\mathbf{v}\|) + 2\epsilon i M^i\|\mathbf{v}\|$$

$$\leq \epsilon M^i(2i + 1 + \frac{2\epsilon i}{M})\|\mathbf{v}\|$$

$$\leq 2\epsilon(i+1)M^i\|\mathbf{v}\|,$$

where the last inequality uses the condition $2\epsilon i \leq 2\epsilon k \leq M$. $\square$

### D.1   PCP Solver

Given a squared ridge regression solver $\texttt{RidgeSquare}(\mathbf{A}, \lambda, c_{2i-1}, \mathbf{v}, \epsilon, \delta)$ (see Section 3), using the reduction in Section 2 we can get an $\epsilon$-approximate PCP algorithm $\texttt{ISPCP}(\mathbf{A}, \mathbf{v}, \lambda, \gamma, \epsilon, \delta)$ shown in Algorithm 5 and its theoretical guarantee in Theorem 1.

---

**Algorithm 5:** $\texttt{ISPCP}(\mathbf{A}, \mathbf{v}, \lambda, \gamma, \epsilon, \delta)$

---

**Input:** $\mathbf{A}$ data matrix, $\mathbf{v}$ projecting vector, $\lambda$ threshold, $\gamma$ eigengap, $\epsilon$ accuracy, $\delta$ probability.
**Parameter:** degree $k$ (Theorem 5), coefficients $\{c_i\}_{i=1}^{2k}, C$ (Eq. (5)), accuracy $\epsilon_1$ (specified
        below)
**Output:** A vector $\widetilde{\mathbf{v}}$ that solves PCP $\epsilon$-approximately.

1  $\widetilde{\mathbf{v}} \leftarrow \mathbf{v}$
2  **for** $i \leftarrow 1$ *to* $k$ **do**
3  $\quad$ $\widetilde{\mathbf{v}} \leftarrow (\mathbf{A}^\top\mathbf{A} - \lambda\mathbf{I})^2\widetilde{\mathbf{v}} + c_{2i}\widetilde{\mathbf{v}}$
4  $\quad$ $\widetilde{\mathbf{v}} \leftarrow \texttt{RidgeSquare}(\mathbf{A}, \lambda, c_{2i-1}, \widetilde{\mathbf{v}}, \epsilon_1, \delta/k)$
5  $\widetilde{\mathbf{v}} \leftarrow C(\mathbf{A}^\top\mathbf{A} - \lambda\mathbf{I})\widetilde{\mathbf{v}}, \widetilde{\mathbf{v}} \leftarrow \frac{1}{2}(\mathbf{v} + \widetilde{\mathbf{v}})$.

---

*Proof of Theorem 1.*

**Choice of parameters:** We choose the following values for parameters in Algorithm 5:

$$k = \Omega(\log(1/\epsilon)\log(1/\lambda\gamma))$$
$$M = \beta_3 k^4 / \beta_2 \gamma^2 \lambda^2$$
$$\epsilon_1 = \frac{\epsilon}{8\beta_3 k^3 M^{k-1}}.$$

The other coefficients $\{c_i\}_{i=1}^{2k}, C$ are as defined in Eq. (5). Further we use constants $\beta_2, \beta_3$ as stated in Lemma 5.

**Approximation:** Given $\lambda > 0, \gamma \in (0,1)$, from Theorem 5 and the definition of $k$ and $r_k^{\lambda\gamma}(x)$ we get that $\max_{|x| \in [\lambda\gamma, 1]} |\mathrm{sgn}(x) - r_k^{\lambda\gamma}(x)| \leq 2\epsilon$.

Using Lemma 1, we know for such $r_k^{\lambda\gamma}$, $\widetilde{\mathbf{v}} = \frac{1}{2}(r_k^{\lambda\gamma}(\mathbf{A}^\top \mathbf{A} - \lambda\mathbf{I}) + \mathbf{I})\mathbf{v}$ satisfies the conditions in (1), i.e.

1. $\|\mathbf{P}_{(1+\gamma)\lambda}(\widetilde{\mathbf{v}} - \mathbf{v})\| \leq \epsilon/2\|\mathbf{v}\|$;
2. $\|(\mathbf{I} - \mathbf{P}_{(1-\gamma)\lambda})\widetilde{\mathbf{v}}\| \leq \epsilon/2\|\mathbf{v}\|$;
3. $\|(\mathbf{P}_{(1+\gamma)} - \mathbf{P}_{(1-\gamma)\lambda})(\widetilde{\mathbf{v}} - \mathbf{v})\| \leq \|(\mathbf{P}_{(1+\gamma)} - \mathbf{P}_{(1-\gamma)\lambda})\mathbf{v}\|$.

Now if we have an approximate solution $\widetilde{\mathbf{v}}'$ satisfying $\|\widetilde{\mathbf{v}}' - \widetilde{\mathbf{v}}\| \leq \epsilon/2\|\mathbf{v}\|$, we check the three conditions respectively. For the first condition we have

$$
\begin{aligned}
\|\mathbf{P}_{(1+\gamma)\lambda}(\widetilde{\mathbf{v}}' - \mathbf{v})\| &\leq \|\mathbf{P}_{(1+\gamma)\lambda}(\widetilde{\mathbf{v}}' - \widetilde{\mathbf{v}})\| + \|\mathbf{P}_{(1+\gamma)\lambda}(\widetilde{\mathbf{v}} - \mathbf{v})\| \\
&\leq \|\widetilde{\mathbf{v}}' - \widetilde{\mathbf{v}}\| + \epsilon/2\|\mathbf{v}\| \\
&\leq \epsilon/2\|\mathbf{v}\| + \epsilon/2\|\mathbf{v}\| \\
&\leq \epsilon\|\mathbf{v}\|,
\end{aligned}
$$

while the third inequality uses the fact that $\mathbf{P}_{(1+\gamma)\lambda}$ is a projection matrix.

For the second condition, we have

$$
\begin{aligned}
\|(\mathbf{I} - \mathbf{P}_{(1-\gamma)\lambda})\widetilde{\mathbf{v}}'\| &\leq \|(\mathbf{I} - \mathbf{P}_{(1-\gamma)\lambda})(\widetilde{\mathbf{v}}' - \widetilde{\mathbf{v}})\| + \|(\mathbf{I} - \mathbf{P}_{(1-\gamma)\lambda})\widetilde{\mathbf{v}}\| \\
&\leq \|\widetilde{\mathbf{v}}' - \widetilde{\mathbf{v}}\| + \epsilon/2\|\mathbf{v}\| \\
&\leq \epsilon/2\|\mathbf{v}\| + \epsilon/2\|\mathbf{v}\| \\
&\leq \epsilon\|\mathbf{v}\|,
\end{aligned}
$$

where for the second inequality we use the fact that $\mathbf{I} - \mathbf{P}_{(1-\gamma)\lambda}$ is also a projection matrix when $\mathbf{P}_{(1-\gamma)\lambda}$ is a projection matrix.

For the last condition, we have

$$
\begin{aligned}
\|(\mathbf{P}_{(1+\gamma)} - \mathbf{P}_{(1-\gamma)\lambda})(\widetilde{\mathbf{v}}' - \mathbf{v})\| &\leq \|(\mathbf{P}_{(1+\gamma)} - \mathbf{P}_{(1-\gamma)\lambda})(\widetilde{\mathbf{v}}' - \widetilde{\mathbf{v}})\| + \|(\mathbf{P}_{(1+\gamma)} - \mathbf{P}_{(1-\gamma)\lambda})(\widetilde{\mathbf{v}} - \mathbf{v})\| \\
&\leq \|\widetilde{\mathbf{v}}' - \widetilde{\mathbf{v}}\| + \|(\mathbf{P}_{(1+\gamma)} - \mathbf{P}_{(1-\gamma)\lambda})\mathbf{v}\| \\
&\leq \|(\mathbf{P}_{(1+\gamma)} - \mathbf{P}_{(1-\gamma)\lambda})\mathbf{v}\| + \epsilon\|\mathbf{v}\|,
\end{aligned}
$$

where for the second inequality we use the fact that $\mathbf{P}_{(1+\gamma)} - \mathbf{P}_{(1-\gamma)\lambda}$ is a projection matrix.

Consequently, it suffices to have such $\widetilde{\mathbf{v}}'$ that $\|\widetilde{\mathbf{v}}' - \widetilde{\mathbf{v}}\| \leq \epsilon/2\|\mathbf{v}\|$ when $\widetilde{\mathbf{v}} = \frac{1}{2}(r_k^{\lambda\gamma}(\mathbf{A}^\top \mathbf{A} - \lambda\mathbf{I}) + \mathbf{I})\mathbf{v}$, note that

$$r_k^{\lambda\gamma}((\mathbf{A}^\top \mathbf{A} - \lambda\mathbf{I}))\mathbf{v} = C(\mathbf{A}^\top \mathbf{A} - \lambda\mathbf{I}) \prod_{i=1}^{k} \frac{(\mathbf{A}^\top \mathbf{A} - \lambda\mathbf{I})^2 + c_{2i}\mathbf{I}}{(\mathbf{A}^\top \mathbf{A} - \lambda\mathbf{I})^2 + c_{2i-1}\mathbf{I}}\mathbf{v}.$$

Suppose we have a procedure $\mathcal{C}_i(\mathbf{v}), i \in [k]$ that can apply $\mathbf{C}_i\mathbf{v}$ for arbitrary $\mathbf{v}$ to $\epsilon'$-multiplicative accuracy with probability $\geq 1 - \delta/k$, where here

$$\mathbf{C}_i = \frac{(\mathbf{A}^\top \mathbf{A} - \lambda\mathbf{I})^2 + c_{2i}\mathbf{I}}{(\mathbf{A}^\top \mathbf{A} - \lambda\mathbf{I})^2 + c_{2i-1}\mathbf{I}}.$$

Also we assume matrix vector product is accurate without loss of generality.[3] Note that

$$\left\|\frac{(\mathbf{A}^\top \mathbf{A} - \lambda \mathbf{I})^2 + c_{2i}\mathbf{I}}{(\mathbf{A}^\top \mathbf{A} - \lambda \mathbf{I})^2 + c_{2i-1}\mathbf{I}}\right\| \leq M, \forall i \in [k],$$

with $M = \beta_3 k^4 / \beta_2 \gamma^2 \lambda^2$. Here we use constants $\beta_2, \beta_3$ as stated in Lemma 5. Now we can use Lemma 12 with the corresponding $M$ to show that

Using a union bound, with probability $\geq 1 - \delta$ it holds that

$$\|\mathcal{C}_k(\mathcal{C}_{k-1}(\cdots \mathcal{C}_1(\mathbf{v}))) - \prod_{i=1}^{k} \mathbf{C}_i \mathbf{v}\| \leq 2\epsilon' k M^{k-1} \|\mathbf{v}\|,$$

whenever $\epsilon' \leq M/(2k)$.

Now we choose

$$\tilde{\epsilon}_1 = \min(\frac{M}{2k}, \frac{\epsilon}{8kM^{k-1}}), \quad \epsilon_1 = \min(\frac{M}{2k}, \frac{\epsilon}{8kM^{k-1}})/(\beta_3 k^2) = \frac{\epsilon}{8\beta_3 k^3 M^{k-1}},$$

consider the following procedure as in Algorithm 5,

$$\mathbf{v} \leftarrow \texttt{RidgeSquare}\big(\mathbf{A}, \lambda, c_{2i-1}, (\mathbf{A}^\top \mathbf{A} - \lambda \mathbf{I})^2 \mathbf{v} + c_{2i}\mathbf{v}, \epsilon_1, \delta/k\big); \forall i \in [k].$$
$$\mathbf{v} \leftarrow C(\mathbf{A}^\top \mathbf{A} - \lambda \mathbf{I})\mathbf{v}.$$

The above choice of $\epsilon_1$ guarantees $\texttt{RidgeSquare}\big(\mathbf{A}, \lambda, c_{2i-1}, (\mathbf{A}^\top \mathbf{A} - \lambda \mathbf{I})^2 \mathbf{v} + c_{2i}\mathbf{v}, \epsilon_1, \delta/k\big)$ for all $i \in [k]$ can be abstracted as $\mathcal{C}_i(\mathbf{v})$ with $\tilde{\epsilon}_1$-accuracy and corresponding success probability. Using a union bound or successful events and also the fact that $\|C(\mathbf{A}^\top \mathbf{A} - \lambda \mathbf{I})\| \leq 2$, we can argue that with probability $\geq 1 - \delta$, the output $\widetilde{\mathbf{v}}'$ of $\texttt{ISPCP}(\mathbf{A}, \mathbf{v}, \lambda, \gamma, \epsilon, \delta)$ satisfy

$$\|\widetilde{\mathbf{v}}' - \widetilde{\mathbf{v}}\| \leq 4\tilde{\epsilon}_1 k M^{k-1} \|\mathbf{v}\| \leq \epsilon/2 \|\mathbf{v}\|.$$

As a result, the output of the algorithm satisfies conditions (1) as desired.

**Runtime:** The numerical constants $C, \{c_i\}_{i=1}^{2k}$ are precomputed. So the runtime will then be a total runtime of computing matrix vector products for $2k+1$ times, calling $k = O(\log(1/\epsilon)\log(1/\lambda\gamma))$ times $\texttt{RidgeSquare}\big(\mathbf{A}, \lambda, c_{2i-1}, (\mathbf{A}^\top \mathbf{A} - \lambda \mathbf{I})^2 \mathbf{v} + c_{2i}\mathbf{v}, \epsilon_1, \delta/k\big)$ for $i \in [k]$. We bound the two terms respectively.

Computing matrix vector products takes time $O(k \text{nnz}(\mathbf{A})) = \tilde{O}(\text{nnz}(\mathbf{A}))$ since $k = \tilde{O}(1)$.

Using Theorem 3, since $\log(1/\epsilon_1) = O(\log(1/\epsilon) + \log k + k\log(M)) = O(\log(1/\epsilon) + k) = \tilde{O}(1)$, the total runtime for solving squared systems is $\tilde{O}(k(\text{nnz}(\mathbf{A}) + d \cdot \text{sr}(\mathbf{A})/(\gamma^2\lambda^2)))$. Further, it can be accelerated to $\tilde{O}(k\sqrt{\text{nnz}(\mathbf{A})d \cdot \text{sr}(\mathbf{A})}/(\lambda\gamma))$ when $\text{nnz}(\mathbf{A}) \leq d \cdot \text{sr}(\mathbf{A})/(\gamma^2\lambda^2)$.

Combining these bounds it gives a running time of Algorithm 5 of

$$\tilde{O}(\text{nnz}(\mathbf{A}) + d \cdot \text{sr}(\mathbf{A})\frac{1}{\gamma^2\lambda^2}).$$

When $\text{nnz}(\mathbf{A}) \leq d \cdot \text{sr}(\mathbf{A})/(\gamma^2\lambda^2)$, it can be accelerated to $\tilde{O}(\sqrt{\text{nnz}(\mathbf{A})d \cdot \text{sr}(\mathbf{A})}/(\lambda\gamma))$. $\quad\square$

Since we assume $\lambda_1 \in [1/2, 1]$ here, $\kappa = 1/\lambda$. We can write it as

$$\tilde{O}\big(\text{nnz}(\mathbf{A}) + \sqrt{\text{nnz}(\mathbf{A}) \cdot d \cdot \text{sr}(\mathbf{A})}\kappa/\gamma\big)$$

by noticing the preprocessing for $\mathbf{A}$ is just setting $\lambda \leftarrow \Theta(\lambda/\lambda_1)$.

## D.2 PCR Solver

Previous work as shown that solving PCR can be reduced to solving PCP together with ridge regression solver. This reduction was first proposed in Frostig et al. [7] and used in subsequent work [8]. The idea is to first compute $\mathbf{v}^* = \mathbf{P}_\lambda(\mathbf{A}^\top \mathbf{b})$ using PCP and then apply $(\mathbf{A}^\top \mathbf{A})^\dagger \mathbf{v}^*$ stably by some polynomial approximation. More specifically, it is achieved through the following procedure.

$$
\begin{aligned}
\mathbf{s}_0 &\leftarrow \mathcal{A}_{\mathrm{PCP}}(\mathbf{A}^\top \mathbf{b}) \\
\mathbf{s}_1 &\leftarrow \texttt{RidgeReg}(\mathbf{A}, \lambda, \mathbf{s}_0), \ \forall m = 1, 2, \cdots, k-1; \\
\mathbf{s}_{m+1} &\leftarrow \mathbf{s}_1 + \lambda \cdot \texttt{RidgeReg}(\mathbf{A}, \lambda, \mathbf{s}_m).
\end{aligned}
\tag{22}
$$

Here `RidgeReg` is a Ridge Regression Solver defined in Definition 5 and `ISPCP` is the $\epsilon$-approximate PCP algorithm specified in Algorithm 5. Such a reduction enjoys the following guarantee:

**Lemma 13** (Reduction: from PCR to PCP)**.** *For fixed $\lambda$, $\epsilon \in (0,1)$ and $\gamma \in (0, 2/3]$, and $\mathbf{A}$ with singular values no more than 1, given an $O(\epsilon/k^2)$-approximate ridge regression solver `RidgeReg`$(\mathbf{A}, \lambda, \cdot)$ and an $O(\gamma\epsilon/k^2)$-approximate PCP solver $\mathcal{A}_{\mathrm{PCP}}(\cdot)$. Running the procedure for $k = \Theta(\log(1/\epsilon\gamma))$ iterations and outputting the final $\mathbf{s}_k$ gives an $\epsilon$-approximate PCR algorithm.*

For completeness, we give the following Algorithm 6 for $\epsilon$-approximate PCP solver and give the proof for its theoretical guarantee as stated in Theorem 2.

---

**Algorithm 6:** `ISPCR`$(\mathbf{A}, \mathbf{b}, \lambda, \gamma, \epsilon, \delta)$

---

**Input:** $\mathbf{A} \in \mathbb{R}^{n \times d}$ properly rescaled, $\mathbf{b} \in \mathbb{R}^d$ regressand, $\lambda > 0$ eigenvalue threshold, $\gamma \in (0, 2/3]$ unitless eigengap, $\epsilon$ desired accuracy, $\delta$ probability parameter.
**Output:** A vector $\mathbf{x} \in \mathbb{R}^n$ that solves PCR $\epsilon$-approximately.

1 Set $k \leftarrow \Theta(\log(1/\epsilon\gamma))$, $\epsilon_1 \leftarrow O(\frac{\gamma\epsilon}{k^2})$, $\epsilon_2 \leftarrow O(\frac{\epsilon}{k^2})$
2 $\mathbf{x} \leftarrow \texttt{ISPCP}(\mathbf{A}, \mathbf{A}^\top \mathbf{b}, \lambda, \gamma, \epsilon_1, \delta/4)$
3 $\mathbf{x}_0 \leftarrow \texttt{RidgeReg}(\mathbf{A}, \lambda, \mathbf{x}, \epsilon_2, \delta/4)$
4 **for** $i \leftarrow 1$ *to* $k-1$ **do**
5 $\quad$ Compute $\mathbf{x} = \lambda \cdot \texttt{RidgeReg}(\mathbf{A}, \lambda, \mathbf{x}, \epsilon_2, \delta/2(k-1)) + \mathbf{x}_0$

---

*Proof of Theorem 2.*

**Approximation:** It follows directly from Lemma 13.

**Runtime:** The total running time consists of one call of `ISPCP` and $k = \Theta(\log(1/\epsilon\gamma))$ calls of `RidgeReg`, with particular parameters specified in Algorithm 6 this leads to a runtime

$$
\tilde{O}(\mathrm{nnz}(\mathbf{A}) + d \cdot \mathrm{sr}(\mathbf{A}) \frac{1}{\gamma^2 \lambda^2}) + k \cdot \tilde{O}(\mathrm{nnz}(\mathbf{A}) + d \cdot \mathrm{sr}(\mathbf{A}) \frac{1}{\lambda}) = \tilde{O}(\mathrm{nnz}(\mathbf{A}) + d \cdot \mathrm{sr}(\mathbf{A}) \frac{1}{\gamma^2 \lambda^2})),
$$

and an accelerated runtime

$$
\tilde{O}(\frac{1}{\gamma\lambda} \sqrt{\mathrm{nnz}(\mathbf{A}) d \cdot \mathrm{sr}(\mathbf{A})})
$$

when $\mathrm{nnz}(\mathbf{A}) \leq d \cdot \mathrm{sr}(\mathbf{A})/(\gamma\lambda)^2$, which proves the result by noticing $\kappa = 1/\lambda$ for rescaled $\mathbf{A}$.

$\square$

## D.3 Square-root Computation

Here we prove Theorem 4. The approach is very similar for developing the PCP solver as in Appendix D.1. We first introduce a rational function that when applied to any given PSD matrix $\mathbf{M}$ well approximates the square-root of itself for matrix $\mu\mathbf{I} \preceq \mathbf{M} \preceq \mathbf{I}$. Note this will immediately generalize to the case when $\mu\mathbf{I} \preceq \mathbf{M} \preceq \lambda\mathbf{I}$ by first getting a constant approximation $\tilde{\lambda}$ of $\lambda$ in $O(\mathrm{nnz}(\mathbf{M}))$ time and then preprocessing $\mathbf{M} \leftarrow \mathbf{M}/\tilde{\lambda}$.

**Lemma 14.** *Given any $\mu\mathbf{I} \preceq \mathbf{M} \preceq \mathbf{I}$ and $\epsilon \in (0,1)$, there is a rational function $r(x)$ with degree $k = O(\log(1/\epsilon)\log(1/\mu))$ satisfying*

$$
\|(r(\mathbf{M}) - \mathbf{M}^{1/2})\mathbf{v}\| \leq \epsilon \|\mathbf{M}^{1/2}\mathbf{v}\|, \forall \mathbf{v} \in \mathbb{R}^n.
$$

In short, such a rational function can be denoted as $\hat{r}_k^\mu(x)$ and expressed as:

$$\hat{r}_k^\mu(x) = Cx \prod_{i\in[k]} \frac{x+c_{2i}}{x+c_{2i-1}} \text{ with } c_i \overset{\text{def}}{=} \mu \frac{\text{sn}^2(\frac{iK'}{2k+1};\sqrt{\mu'})}{\text{cn}^2(\frac{iK'}{2k+1};\sqrt{\mu'})}, i\in[2k]. \tag{23}$$

$$\text{coefficients} \begin{cases} C & \overset{\text{def}}{=} \frac{2}{(\sqrt{\mu}\prod_{i\in[k]}\frac{\mu+c_{2i}}{\mu+c_{2i-1}})+(\prod_{i\in[k]}\frac{1+c_{2i}}{1+c_{2i-1}})}, \\ c_i & \overset{\text{def}}{=} \mu \frac{\text{sn}^2(\frac{iK'}{2k+1};\sqrt{\mu'})}{\text{cn}^2(\frac{iK'}{2k+1};\sqrt{\mu'})}, \forall i\in\{1,2,\cdots,2k\}. \end{cases}$$

$$\text{numerical constants} \begin{cases} \sqrt{\mu'} & \overset{\text{def}}{=} \sqrt{1-\mu}, \\ K' & \overset{\text{def}}{=} \int_0^{\pi/2} \frac{d\theta}{\sqrt{1-\mu'\sin^2\theta}}, \\ u & \overset{\text{def}}{=} F(\phi;\sqrt{\mu'}) \overset{\text{def}}{=} \int_0^\phi \frac{d\theta}{\sqrt{1-\mu'\sin^2\theta}}, \\ \text{sn}(u;\sqrt{\mu'}) & \overset{\text{def}}{=} \sin(F^{-1}(u;\sqrt{\mu'})); \text{cn}(u;\sqrt{\mu'}) \overset{\text{def}}{=} \cos(F^{-1}(u;\sqrt{\mu'})). \end{cases}$$

It is easy to check that $\hat{r}_k^\mu(x)$ can be related to the rational function defined in Section 2 through formula $\hat{r}_k^\mu(x^2) = x\cdot r_k^{\sqrt{\mu}}(x)$, where $r_k^\gamma(x)$ is defined as in (5) By Theorem 5, with the same order of $k \geq \Omega(\log(1/\epsilon)\log(1/\mu))$. A formal proof also utilizes such relationship between the rational functions.

*Proof.* Consider a modified rational function of Zolotarev rational as follows:

$$\hat{r}_k^\mu(x) = Cx \prod_{i=1}^k \frac{x+c_{2i}}{x+c_{2i-1}} \text{ with } c_i \overset{\text{def}}{=} \mu \frac{\text{sn}^2(\frac{iK'}{2k+1};\sqrt{\mu'})}{\text{cn}^2(\frac{iK'}{2k+1};\sqrt{\mu'})}, i\in[2k], \tag{24}$$

with coefficients $C$ as the corresponding $C$ in $r_k^{\sqrt{\mu}}(x)$, $\sqrt{\mu'} \overset{\text{def}}{=} \sqrt{1-\mu}$ and $K' \overset{\text{def}}{=} \int_0^{\pi/2} \frac{d\theta}{\sqrt{1-\mu'\sin^2\theta}}$, $\text{sn}(u;\sqrt{\mu'}) \overset{\text{def}}{=} \sin(F^{-1}(u;\sqrt{\mu'}))$, $\text{cn}(u;\sqrt{\mu'}) \overset{\text{def}}{=} \cos(F^{-1}(u;\sqrt{\mu'}))$ with definition $u = F(\phi;\sqrt{\mu'}) \overset{\text{def}}{=} \int_0^\phi \frac{d\theta}{\sqrt{1-\mu'\sin^2\theta}}$.

Note this rational actually satisfies the condition that $\hat{r}_k^\mu(x^2) = x\cdot r_k^{\sqrt{\mu}}(x)$. By Theorem 5, it holds that when $k \geq \Omega(\log(1/\epsilon)\log(1/\mu))$

$$|x\cdot r_k^{\sqrt{\mu}}(x) - x\cdot\text{sgn}(x)| \leq \epsilon|x|, \forall|x|\in[\sqrt{\mu},1]. \tag{25}$$

Now if we write $\mathbf{M} = \mathbf{V}\boldsymbol{\Lambda}\mathbf{V}^\top$ where $\boldsymbol{\Lambda} = \text{diag}(\lambda_1,\cdots,\lambda_n)$ satisfying $1 \geq \lambda_1 \geq \cdots, \lambda_n \geq \mu$ and that each column of $\mathbf{V}$ is $\boldsymbol{\nu}_i, i\in[n]$. We can write $\mathbf{v} = \sum_{i=1}^n \alpha_i\boldsymbol{\nu}_i$, and thus get $\mathbf{M}^{1/2}\mathbf{v} = \sum_{i=1}^n \alpha_i\sqrt{\lambda_i}\boldsymbol{\nu}_i$.

Now we consider what we can get from substituting $x$ in $\hat{r}_k^\mu(x)$ with $\mathbf{M}$. By applying Eq. (25) and substituting $x^2$ with $\mathbf{M}$, we get

$$\|\hat{r}_k^\mu(\mathbf{M})\mathbf{v} - \sqrt{\mathbf{M}}\mathbf{v}\| \leq \epsilon\|\mathbf{M}^{1/2}\mathbf{v}\|, \forall\mathbf{v}\in\mathbb{R}^n.$$

$\square$

Now we give the following pseudocode in Algorithm 7 for $\texttt{SquareRoot}(\mathbf{M},\mathbf{v},\epsilon,\delta)$, which with probability $1-\delta$ outputs a solution $\mathbf{x}$ satisfying $\|\mathbf{x} - \mathbf{M}^{1/2}\mathbf{v}\| \leq \epsilon\|\mathbf{M}^{1/2}\mathbf{v}\|$. In the pseudocode,

we use $\texttt{LinearSolver}(\mathbf{M}, c, \mathbf{x}, \epsilon, \delta)$ to denote any solver that with probability $1 - \delta$ gives as $\epsilon$-approximate solution $\tilde{\mathbf{x}}$ of $\mathbf{x}^* = (\mathbf{M} + cI)^{-1}\mathbf{v}$ satisfying $\|\tilde{\mathbf{x}} - \mathbf{x}^*\| \le \epsilon\|\mathbf{v}\|$.

---

**Algorithm 7:** $\texttt{SquareRoot}(\mathbf{M}, \mathbf{v}, \epsilon, \delta)$

---

**Input:** $\mathbf{M} \in \mathbb{R}^{n \times n}$ data matrix, $\mathbf{v} \in \mathbb{R}^n$ vector, $\epsilon$ accuracy, $\delta$ probability.
**Parameter:** Smallest eigenvalue $\mu$, largest eigenvalue $\lambda$, degree $k$ (Lemma 14), coefficients
$\qquad \{c_i\}_{i=1}^{2k}, C$ (Eq. (24)), accuracy $\epsilon_1$ (specified below)
**Output:** A vector $\mathbf{x}$ satisfying $\|\mathbf{x} - \mathbf{M}^{1/2}\mathbf{v}\| \le \epsilon\|\mathbf{M}^{1/2}\mathbf{v}\|$.

1 $\mathbf{x} \leftarrow \mathbf{v}$
2 **for** $i \leftarrow 1$ *to* $k$ **do**
3 $\quad\quad \mathbf{x} \leftarrow \mathbf{M}\mathbf{x} + c_{2i}\mathbf{x}$
4 $\quad\quad \mathbf{x} \leftarrow \texttt{LinearSolver}(\mathbf{M}, c_{2i-1}, \mathbf{x}, \epsilon_1, \delta/k)$
5 $\mathbf{x} \leftarrow C \cdot \mathbf{M}\mathbf{x}$.

---

*Proof of Theorem 4.*

Without loss of generality, we can assume $\lambda = 1$, otherwise one can consider $\mathbf{M}/\lambda$ instead.

**Choice of parameters:** We choose the following values for parameters in Algorithm 7:

$$k = \Omega(\log(1/\epsilon)\log(1/\mu))$$
$$M = \beta_3 k^4 / \beta_2 \mu$$
$$\epsilon_1 = \frac{\epsilon}{8\beta_3 k^3 M^{k-1}}.$$

Other coefficients $\{c_i\}_{i=1}^{2k}, C$ are as defined in Eq. (5). Here we use constants $\beta_2, \beta_3$ as stated in Lemma 5.

**Approximation:** Given $\epsilon > 0, \mu > 0$, from Lemma 14 we set $k \ge \Omega(\log(1/\epsilon)\log(1/\mu))$ thus $\hat{r}_k^\mu(x)$ as defined in Eq. (24) satisfies $\|\mathbf{M}^{1/2}\mathbf{v} - r_k^\mu(\mathbf{M})\mathbf{v}\| \le \epsilon/2\|\mathbf{M}^{1/2}\mathbf{v}\|$. Now it suffices to get a $\mathbf{x}$ satisfying $\|\mathbf{x} - r_k^\mu(\mathbf{M})\mathbf{v}\| \le \epsilon/2\|\mathbf{M}^{1/2}\mathbf{v}\|$.

Suppose we have a procedure $\mathcal{C}_i(\mathbf{v}), i \in [k]$ that can apply $\mathbf{C}_i\mathbf{v}$ for arbitrary $\mathbf{v}$ to $\epsilon'$-multiplicative accuracy with probability $\ge 1 - \delta/k$, where here

$$\mathbf{C}_i = \frac{\mathbf{M} + c_{2i}\mathbf{I}}{\mathbf{M} + c_{2i-1}\mathbf{I}}.$$

Also we assume matrix vector product is accurate without loss of generality.[4] Note that

$$\left\|\frac{\mathbf{M} + c_{2i}\mathbf{I}}{\mathbf{M} + c_{2i-1}\mathbf{I}}\right\| \le M, \forall i \in [k],$$

with $M = \beta_3 k^4 / \beta_2 \mu$. Here we use constants $\beta_2, \beta_3$ as stated in Lemma 5. Now we can use Lemma 12 with the corresponding $M$ to show that: Using a union bound, with probability $\ge 1 - \delta$ it holds that

$$\left\|\mathcal{C}_k(\mathcal{C}_{k-1}(\cdots\mathcal{C}_1(\mathbf{v}))) - \prod_{i=1}^k \mathbf{C}_i\mathbf{v}\right\| \le 2\epsilon'kM^{k-1}\|\mathbf{v}\|,$$

whenever $\epsilon' \le \frac{M}{2k}$.

Now we choose

$$\tilde{\epsilon}_1 = \min\left(\frac{M}{2k}, \frac{\sqrt{\mu}\epsilon}{8kM^{k-1}}\right), \quad \epsilon_1 = \min\left(\frac{M}{2k}, \frac{\sqrt{\mu}\epsilon}{8kM^{k-1}}\right)/(\beta_3 k^2) = \frac{\epsilon\sqrt{\mu}}{8\beta_3 k^3 M^{k-1}},$$

consider the following procedure as in Algorithm 5,

$$\mathbf{x}\leftarrow\texttt{LinearSolver}(\mathbf{M},c_{2i-1},\mathbf{x},\epsilon_1,\delta/k);\forall i\in[k].$$
$$\mathbf{x}\leftarrow C\cdot\mathbf{M}\mathbf{x}.$$

The above choice of $\epsilon_1$ guarantees procedures $\texttt{LinearSolver}(\mathbf{M},c_{2i-1},\mathbf{x},\epsilon_1,\delta/k)$ for all $i\in[k]$ can be abstracted as $\mathcal{C}_i(\mathbf{v})$ with $\tilde{\epsilon}_1$-accuracy and corresponding success probability. Using a union bound of successful events and the fact that $\|C\cdot\mathbf{M}\|\leq2$, we can argue that with probability $1-\delta$, the output $\mathbf{x}$ of $\texttt{SquareRoot}(\mathbf{M},\mathbf{v},\epsilon,\delta)$ satisfy

$$\|\mathbf{x}-r_k^\mu(\mathbf{M})\mathbf{x}\|\leq4\tilde{\epsilon}_1kM^{k-1}\|\mathbf{v}\|\leq\frac{4\tilde{\epsilon}_1kM^{k-1}}{\sqrt{\mu}}\|\mathbf{M}^{1/2}\mathbf{v}\|\leq\epsilon/2\|\mathbf{M}^{1/2}\mathbf{v}\|,$$

By Definition 4, using the above choice of parameters we conclude this justifies the correctness argument in the theorem.

**Runtime:** Given the The numerical constants $C,\{c_i\}_{i=1}^{2k}$ are precomputed. So the runtime will then be a total runtime of computing matrix vector products for $k+1$ times, calling $k=O(\log(1/\epsilon)\log(1/\mu))$ times $\texttt{LinearSolver}(\mathbf{M},c_{2i-1},\mathbf{x},\epsilon_1,\delta/k)$ for $i\in[k]$. We bound the two terms respectively.

Computing matrix vector products takes time $\tilde{O}(\text{nnz}(\mathbf{M}))$ since $k=\tilde{O}(1)$.

Running time of $\texttt{LinearSolver}(\mathbf{M},c_{2i-1},\mathbf{x},\epsilon_1,\delta/k)$ depends on the particular solver we use. Based on the assumption and the fact that $c_{2i-1}\in[\tilde{\Omega}(\mu),\tilde{O}(1)],\forall i\in[k]$, we can upper bound each solve by $\tilde{O}(\mathcal{T})$.

Adding it together gives running time of Algorithm 7 in

$$\tilde{O}(\text{nnz}(\mathbf{M})+\mathcal{T}).$$

Replacing $\mathbf{M}$ with $\mathbf{M}/\lambda$, $\mu$ above with $\mu/\lambda$ and $\lambda$ with 1 due to preprocessing gives the final statement. $\qquad\square$

# E    More on Experiments

In this section we give a more detailed description and theoretical justification for *rlanczos* and *slanczos*. Also we show its relationship and difference with our proposed algorithm ISPCP.

## E.1    Details for *rlanczos*

Lanczos method is well known for being efficient and stable [17]. As verified in theory and practice, running Lanczos algorithm on $(\mathbf{A}^\top\mathbf{A}+\lambda\mathbf{I})^{-1}(\mathbf{A}^\top\mathbf{A}-\lambda\mathbf{I})$ almost always beats the optimal universal approximation of $\text{sgn}(x)$ by stably applying polynomial / chebyshev. This is because Lanczos can search for the best polynomial to approximate $\text{sgn}(x)$ based on the distribution of $\mathbf{A}^\top\mathbf{A}$'s eigenvalues.

Based on that, we also combined the well-studied rational Lanczos algorithm [21] with the known Zolotarev rational expression, to search in the rational function space $r_{2k+1}(x)\in\mathcal{P}_{2k+1}/q_{2k}(x)$ where $q_{2k}(x)$ has exactly expression as in the denominator of $r_k^\gamma(x)$. Theoretically, this should always beat directly applying Zolotarev rational by allowing more freedom to cater to the distribution of eigenvalues of $\mathbf{A}^\top\mathbf{A}$.

The two methods *rational* and *rlanczos* are quite close in practice. (see Figs. 2 and 3) In general for low accuracy regime, *rlanczos* slightly improve on *rational* ISPCP. But ISPCP is more stable and can get to more accurate solutions. This also shows the strength of ISPCP proposed in the paper.

## E.2    Details for *slanczos*

For *slanczos*, the idea is to incorporate the squared system primitive with lanczos method on polynomial directly, i.e. searching for function in form

$$f\left(\frac{(x-\lambda)(x+\lambda)}{(x-\lambda)^2+\gamma(x+\lambda)^2}\right)$$

and replace $x \leftarrow \mathbf{A}^\top \mathbf{A}$.

Note here we introduce shift-and-rescaling $(\mathbf{A}^\top \mathbf{A} + \lambda \mathbf{I})^{-1}(\mathbf{A}^\top \mathbf{A} - \lambda \mathbf{I})$ so that all eigenvalues of $\mathbf{A}^\top \mathbf{A}$ satisfying $\lambda_i \in [0, \lambda(1-\gamma) \cup (\lambda(1+\gamma), \infty)$ are mapped to range $[-1, -\gamma/2] \cup [\gamma/2, 1]$. So for now let's consider $|x| \in [\gamma/2, 1]$. One observation is that there is this squared primitive in form $x/(x^2 + \gamma)$ can map $|x| \in [\gamma/2, 1]$ to $[\Theta(1), \Theta(1/\sqrt{\gamma})]$.

**Lemma 15.** *Take* $r(x) = x/(x^2 + \gamma)$. *Then* $\tilde{r}(x) = 2\sqrt{\gamma} r(x)$ *maps* $x \in [\gamma/2, 1]$ *to* $(2\sqrt{\gamma}/3, 1)$ *and* $x \in [-1, -\gamma/2]$ *to* $(-1, -2\sqrt{\gamma}/3)$.

*Proof.* We only consider $x \in [\gamma/2, 1]$ and a exactly symmetric argument works for the other side. Now consider $1/r(x) = x + \gamma/x$, we have when $x \in [\gamma/2, 1]$, $1/r(x) \in [2\sqrt{\gamma}, 2 + \gamma]$, thus showing $r(x) \in [1/(2+\gamma), 1/(2\sqrt{\gamma})] \subseteq (1/3, 1/2\sqrt{\gamma})$. Multiplying by coefficient we have $\tilde{r}(x) \in (2\sqrt{\gamma}/3, 1)$ whenever $x \in [\gamma/2, 1]$, which completes the proof. $\square$

**Remark 1.** *For such a rational primitive* $\tilde{r}(x)$ *since we know there is optimal* $O(\log(1/\epsilon)/\sqrt{\gamma})$-*degree chebyshev polynomial* $f(x)$ *that maps* $[\sqrt{\gamma}, 1]$ *to* $[1 - \epsilon, 1]$ *and* $[-1, -\sqrt{\gamma}]$ *to* $[-1, -1 + \epsilon]$, *thus we are able to run only* $\tilde{O}(1/\sqrt{\gamma})$ *suboracles of ridge regression of solving* $((\mathbf{A}^\top \mathbf{A} - \lambda \mathbf{I})^2 + \gamma(\mathbf{A}^\top \mathbf{A} + \lambda \mathbf{I}))\mathbf{x} = \mathbf{v}$.

Formally combining this with the squared system solver we develop in Theorem 3 leads to the following theoretical guarantee:

**Theorem 12** (Runtime for *slanczos*)**.** slanczos *can be converted into an* $\epsilon$-*approximate PCP / PCR solver with runtime guarantee*

$$\tilde{O}\left(\mathrm{nnz}(\mathbf{A})/\sqrt{\gamma} + \sqrt{\mathrm{nnz}(\mathbf{A})d \cdot \mathrm{sr}(\mathbf{A})}\kappa/\gamma\right).$$

This guarantee implies the method *slanczos* would work better than both *polynomial*, *chebyshev* in [7, 8], and rational methods including ISPCP and *rlanczos* (see Appendix E.1) for certain regimes of parameters.

As the runtime of *slanczos* is not almost linear, we don't state it formally in the main part of paper. Also, as $\mathrm{nnz}(\mathbf{A})/d \cdot \mathrm{sr}(\mathbf{A})^2 \kappa^2$ or simply $n/d$ gets larger, it gets worse compared with the two purely rational methods. This also is verified by our experiments shown in Fig. 3 - when we fix $d, \gamma$ and increase the magnitude of $n$, *rational* and *rlanczos* start outperforming *slanczos*.

## Footnotes

[3]If in the finite-precision world, we assume arithmetic operations are carried out with $\Omega(\log(n/\epsilon))$ bits of precision, the result is still true by standard argument with a slightly different constant factor for the bounding coefficient.

[4]If in the finite-precision world, we assume arithmetic operations are carried out with $\Omega(\log(n/\epsilon))$ bits of precision, the result is still true by standard argument with a slightly different constant factor for the bounding coefficient.