[Reviews · NeurIPS 2019]

Reviewer 1



The main contribution relies on the AsySVRG to solve the squared ridge regression. This result is interesting and the properties is attractive. However, in the proof, it relies on a key result that $(M+M^T)/2 \succeq \mu I$ implies that $x^TMx \geq \mu x^T x$ (the equation line 475-476). This result is not obvious but detailed proof is missing.

Reviewer 2



## Update ## I'd like to thank the authors for their thorough response. I will maintain my initial score. I did review the discussion of rLanczos and sLanczos in Appendices F.1 and F.2 and agree that several additional sentences in the main paper should suffice to clarify the methods. ## Initial Review ## The authors propose a nearly linear-time randomized algorithm for principle component projection (PCP) and principle component regression (PCR). This is achieved using Zolotarev rational functions to approximate the sign function instead of the polynomial function approximations used in previous work. Computing the Zolotarev rational functions requires solving squared ridge-regression problems. The authors reduce solving squared ridge-regression problems to solving asymmetric linear systems and analyze convergence of a SVRG solver for generic asymmetric linear systems. The properties of the SVRG solver are further analyzed in the context of the specific asymmetric systems corresponding to these squared ridge-regression problems, which yields a tighter convergence (in high-probability) bound. Accelerated high-probability rates are proved in both the generic and specialized settings. The accelerated rate specialized to solving squared ridge-regression problems yields the proposed algorithm for PCP and, by reduction, PCR. The paper concludes with an experimental evaluation on several synthetic PCP problems. Variants of the proposed algorithm are shown to perform faster than methods based-on polynomial approximations and competitively with the Lanczos algorithm. Originality: The proposed algorithms are novel. Related work is appropriately cited. Clarity: The paper is clear and easy to follow. However, I suggest that some additional discussion of rLanczos and sLanczos be added to main text if possible. See minor comments below. Quality and Significance: This paper appears to resolve an significant outstanding problem with respect to randomized PCP and PCR algorithms. As a part of the analysis, several high-probability convergence bounds are proved for SVRG methods applied to solving asymmetric linear systems. This alone seems like an excellent contribution and a step towards faster linear algebra primitives via the modern optimization toolbox. Empirical investigations are provided to confirm the authors' theoretical conclusions. These experiments are small, but well-executed. Minor Comments: - Line 375: The title of Appendix C should read "Proofs for Results in Section 2". - Line 202 should read "We denote ... as *the* cost of *the* matrix-vector of ..." - Line 227 should read "... *the* data matrix A..." - Line 230 should read "In all figures below, *the* y-axis *denotes*..." - Section 2 might flow better if Figure 1 was placed at the top of the page, rather than embedded in the text. The same is true for Algorithm 1 and Figure 3.

Reviewer 3



This paper improves upon existing approaches to the principle component projection problem in an interesting way. The approach builds upon existing approaches by using more sophisticated techniques for matrix sign approximation, and for solving the resulting subproblem that arises. As a reviewer I'm knowledgable on SVRG and variants rather than the PCP problem, so I can't comment much on the novelty of the innovations specific to PCP. The application of SVRG here is interesting however. I'm not aware of previous applications of SVRG to asymmetric matrix solves as done here. They use a weighted sampling variant to get tighter bounds, which is interesting. Weighting sampling doesn't see much practical use as the sampling weights can be hard to determine, however for matrix solves there is a clear choice as used here. The experiments look well executed, however the focus on a single synthetic task is a definite weakness of the paper, it would be nice to see a real-world application. The paper is well written, although very dense. I think the description of the results is presented with sufficient formality. I don't have too many comments in that regard. In terms of significance this seems like a solid improvement in the research area. Smaller notes: Abstract: “had superlinear running times” should be sub-linear? I'm unsure. “directly applied to problem” should be “directly applied to this problem” “PCP and PCR to solving following “ should be “PCP and PCR to solving the following” “This is stated in formal in the” should be “This is stated formally in the” “compared with ones “ should be “compared with the ones “

[Author Response · NeurIPS 2019]

We are glad the reviewers appreciate the theoretical contributions and the claims of the empirical experiments. We remark that these theoretical improvements are the primary contributions of the paper; the experiments were meant primarily to corroborate. We believe that the design of squared / asymmetric linear system solver based on variance reduction is intrinsically interesting. That it provides a new powerful primitive (and alternative to ridge regression) that enables nearly-linear runtimes for solving more complex problems (e.g. PCP / PCR) further demonstrates its utility and is the main result of this paper which the empirical experiments merely corroborate.

**Reviewer 2** Thank you for the review; we are glad you found our result interesting. Below, we address in detail the concern you raised. With this concern addressed, we hope this elevates your view of the paper.

*Proof of statement:* $(M + M^T)/2 \succeq \mu I$ *if and only if* $x^\top M x \geq \mu x^T x$

*Proof.* Since $x^\top M x$ is a scalar by linearity we have

$$x^\top M x = x^\top M^\top x = \frac{1}{2}(x^\top M x + x^\top M^\top x) = x^\top \left( \frac{M^\top + M}{2} \right) x \ .$$

The claim follows as $(M + M^\top)/2 \succeq \mu I$ if and only if $x^\top((M + M^\top)/2)x \geq \mu x^\top x$ for all $x$ by definition of $\succeq$. $\quad\square$

We will be sure to add the proof of this fact in the full version.

**Reviewer 4** Thank you for the kind review and for recognizing the importance of the problem and the novelty of our technique. We do hope both the rational approximation idea and the squared / asymmetric system solver we develop would inspire and assist further development in principle component analysis and linear system solving. We will correct all the typographical problems you mentioned in the final version of the paper.

*Some additional discussion of rLanczos and sLanczos be added to main text if possible.*

Thanks for this kind suggestion! We will add a few sentences giving a high level description of both algorithms. We were unable to demonstrate them in greater detail in the main paper due to the page limit. The particular details of *rLanczos* and *sLanczos* including theoretical runtime and implementation are included in Appendix F.1 and F.2; both methods rely on the key idea of solving squared system as subroutines discussed in Section 3.

**Reviewer 5** Thank you for the kind review; we are pleased that you found the technique interesting. We do view this method for solving certain asymmetric systems as a variant of SVRG, and hope it will inspire and assist future improvement in obtaining faster algorithms on general linear system solving and other optimization tasks. We will address all the minor comments you pointed out in the final version of the paper.

*The experiments look well executed, ..., would be nice to see a real-world application.*

We believe the major contribution of the paper lies in its theoretical side, and we designed experiments on synthetic data primarily to give a clearer view of when and by how much our algorithm works better, confirming the theoretical results. We choose synthetic data because it gives more flexibility in controlling the patterns (eigenvalue distribution, eigen-gap, etc.). We agree that further empirical experiments on real data would be beneficial - it is a very interesting direction that we wish to explore in future work.

*"had superlinear running times" should be sub-linear?*

This is a good question. The previous methods can be superlinear because the dominating term in their runtime are either $k \cdot \text{nnz}(A)$ or $\text{nnz}(A)/\gamma$ where $k$ is the number of eigenvalues above the threshold $\lambda$ and $\gamma$ is the eigen-gap around $\lambda$. If $k$ gets larger or $\gamma$ gets smaller as problem size ($\text{nnz}(A)$) gets bigger then both methods can be superlinear. We will revise the introduction to clarify this in our final version of the paper.

[Meta-Review · NeurIPS 2019]

All reviewers viewed the contribution favourably, and I particular like the asymmetric linear system result.